# Cryo-EM structure of the calcium release-activated calcium channel Orai in an open conformation

Xiaowei Hou[†], Ian R Outhwaite[†], Leanne Pedi[‡], Stephen Barstow Long*

Structural Biology Program, Memorial Sloan Kettering Cancer Center, New York, United States

**Abstract** The calcium release-activated calcium channel Orai regulates $Ca^{2+}$ entry into non-excitable cells and is required for proper immune function. While the channel typically opens following $Ca^{2+}$ release from the endoplasmic reticulum, certain pathologic mutations render the channel constitutively open. Previously, using one such mutation (H206A), we obtained low (6.7 Å) resolution X-ray structural information on *Drosophila melanogaster* Orai in an open conformation (Hou et al., 2018). Here we present a structure of this open conformation at 3.3 Å resolution using fiducial-assisted cryo-electron microscopy. The improved structure reveals the conformations of amino acids in the open pore, which dilates by outward movements of subunits. A ring of phenylalanine residues repositions to expose previously shielded glycine residues to the pore without significant rotational movement of the associated helices. Together with other hydrophobic amino acids, the phenylalanines act as the channel's gate. Structured M1–M2 turrets, not evident previously, form the channel's extracellular entrance.

*For correspondence:
longs@mskcc.org

[†]These authors contributed equally to this work

Present address: [‡]Tri-Institutional Therapeutics Discovery Institute, New York, United States

Competing interests: The authors declare that no competing interests exist.

## Introduction

Intracellular calcium signals in most non-excitable metazoan cells are augmented and shaped by the influx of extracellular $Ca^{2+}$ ions due to opening of the calcium release-activated calcium (CRAC) channel Orai in the plasma membrane (*Hogan et al., 2010*). $Ca^{2+}$ influx through the channel is necessary for activation of immune response genes in T cells and is involved in range of other physiological processes (*Feske et al., 2005*; *Lacruz and Feske, 2015*; *Prakriya and Lewis, 2015*). Through an unusual mechanism, it is the depletion of $Ca^{2+}$ from the endoplasmic reticulum (ER), rather than an increase in the cytosolic $Ca^{2+}$ concentration, that initiates opening of the channel (reviewed in *Hogan et al., 2010*). Although $Ca^{2+}$ entry through the channel is a major contributor to intracellular $Ca^{2+}$ signaling, the physiological functions of the channel are less appreciated than $Ca^{2+}$ release from the ER, partly because the molecular components of the CRAC channel, the Orai protein and its regulator STIM, were identified fairly recently. Orai is an integral membrane protein that forms the pore of the channel in the plasma membrane (*Feske et al., 2006*; *Vig et al., 2006*; *Yeromin et al., 2006*; *Zhang et al., 2006*). There are three Orai proteins in humans (Orai1–3). *Drosophila melanogaster* contains one ortholog (Orai), which shares 73% sequence identity to human Orai1 in the transmembrane region and is the most studied non-human Orai channel. STIM proteins (STIM1 and STIM2 in humans) are single-pass membrane proteins located in the membrane of the ER that regulate Orai channel function in response to $Ca^{2+}$ levels in the ER (*Roos et al., 2005*; *Zhang et al., 2005*). Although the molecular mechanisms of this process are not yet fully resolved, a scheme for channel activation is becoming clear (reviewed in *Krizova et al., 2019*; *Lunz et al., 2019*; *Qiu and Lewis, 2019*). The 'inside-out' signaling from the ER to the plasma membrane occurs at cellular locations where the ER and plasma membranes are close together (separated by ~10–20 nm). $Ca^{2+}$ release from the ER into the cytosol, which can occur via the IP3R receptor, is detected by

a luminal domain of STIM from the reduction of [$Ca^{2+}$] in the ER. An ensuing conformational change in STIM enables its cytosolic domain to interact with Orai across the divide separating the two membranes and instigates opening of the pore of Orai. A high-resolution three-dimensional (3D) structure of Orai in an open conformation would provide molecular context to some of these processes.

Activated CRAC channels have exceedingly low ion conductance in comparison to most other ion channels and they are highly selective for $Ca^{2+}$ (*Lepple-Wienhues and Cahalan, 1996*; *Prakriya and Lewis, 2006*). The unitary conductance of activated CRAC channels is so low (7–25 fS in 2–110 mM $Ca^{2+}$) that recordings of currents from single channels have not been feasible (*Prakriya and Lewis, 2003*; *Prakriya and Lewis, 2006*). Both of these properties, slow ion permeation and high $Ca^{2+}$ selectivity, are fundamental to the channel's ability to generate sustained elevations of cytosolic calcium concentrations, which, among physiological functions, activate immune response genes in T cells (*Hogan et al., 2010*). Mutations in Orai or STIM that cause loss of channel function underlie a spectrum of immunological disorders (*Lacruz and Feske, 2015*). For instance, mutation of a pore-lining residue (R91W) in Orai1 causes a severe combined immune deficiency-like disorder due to lack of functional CRAC channels in the T cells of these patients (*Feske et al., 2006*). In addition to loss-of-function mutations, some gain-of-function mutations have been identified that allow Orai to conduct cations without activation by STIM (reviewed in *Krizova et al., 2019*). Activating mutations of Orai1 have been associated with tubular aggregate myopathy and Stormorken syndromes (*Lacruz and Feske, 2015*). Many of the gain-of-function mutants have reduced ion selectivity for $Ca^{2+}$, as indicated by an altered electrophysiological current–voltage relationship. The H134A mutant of human Orai1, on the other hand, is highly selective for $Ca^{2+}$ and exhibits a similar current–voltage relationship to that of the wild-type channel when it is activated by STIM, which suggests that the pore adopts a similar conformation to the STIM-activated channel (*Frischauf et al., 2017*; *Krizova et al., 2019*; *Yeung et al., 2018*). Biochemical studies, supported by molecular dynamics simulations, also suggest that the conformation of the H134A mutant is highly similar to the naturally opened channel (*Frischauf et al., 2017*; *Yeung et al., 2018*). Unlike many gain-of-function mutations, which occur in amino acids that directly form the walls of the pore, the H134A mutation of Orai1 is located on the M2 helix and not exposed to the pore (*Hou et al., 2018*; *Hou et al., 2012*). We have shown that purified *Drosophila* Orai containing the corresponding H206A mutation (Orai$_{H206A}$) forms a constitutively active channel when reconstituted into liposomes (*Hou et al., 2018*). Orai$_{H206A}$ exhibits properties of the STIM-activated channel, including the ability to conduct $Na^+$ in the absence of divalent cations and the abilities of $Mg^{2+}$ and $Ca^{2+}$ to block the $Na^+$ current (*Hou et al., 2018*). In the absence of structural information for STIM-activated Orai, which has proven difficult to obtain, the H206A mutation provides an experimental approach with which study the three-dimensional (3D) structure of an opened Orai pore.

The X-ray structure of a closed conformation of *Drosophila melanogaster* Orai, referred to as the 'quiescent' conformation (*Hou et al., 2012*), provided the first view of the molecular architecture of the channel. This structure, determined at 3.35 Å resolution, indicated that the channel is formed from a hexamer of Orai subunits rather than a tetrameric assembly, which was anticipated at the time. Subsequent studies have shown that human Orai1 channels also assemble and function as hexamers (*Cai et al., 2016*; *Yen et al., 2016*). The structure revealed that the channel has a single ion conduction pore along a central sixfold axis of symmetry, which would be perpendicular to the membrane in a cellular setting (*Hou et al., 2012*). Each subunit contains four transmembrane helices (M1–M4). Six M1 helices, one from each subunit, form the walls of the ion pore, which is approximately 55 Å long and narrow in the closed conformation. The M2 and M3 helices surround the M1 helices but do not contribute to the pore. The M4 helices are located at the periphery of membrane-spanning portion of the channel while helices extending from them (M4-ext helices) protrude into the cytosol (*Hou et al., 2018*; *Hou et al., 2012*). The linkages between the M1 and M2 helices were not clearly resolved in the electron density and because of this, there was some ambiguity regarding which M1 helix in the hexamer is associated with which M2 helix. The location of the linkages relative to pore suggests that they would contribute to its extracellular entrance, as has been suggested by the effects of mutations of acidic amino acids within them on the binding of pore-blocking lanthanides ($Gd^{3+}$ and $La^{3+}$) (*Yeromin et al., 2006*) and the observation that such mutations

reduce $Ca^{2+}$ influx into cells (*Frischauf et al., 2015*). Structural information regarding the M1–M2 linkage would advance our understanding of the extracellular entrance.

Using the gain-of-function H206A mutation of *Drosophila melanogaster* Orai (Orai$_{H206A}$), we previously determined a low (6.7 Å) resolution X-ray structure of Orai in an open conformation (*Hou et al., 2018*). The structure yielded information regarding the positioning of α-helices in an open conformation of the channel and indicated that opening involves the dilation of the pore. However, the locations of amino acids throughout the channel could only be deduced by comparison with the quiescent structure and the conformations that they adopt in the opened state were not resolved due to the low resolution of the diffraction data (*Hou et al., 2018*). Consequently, the structure provided limited insights into potential changes in the amino acid conformations that occur and the involvement of these changes in the ability of $Ca^{2+}$ to flow through the opened pore.

Structural, functional, and computational studies have suggested that hydrophobic amino acids within the pore (particularly Phe 99 and Val 102 in human Orai1, corresponding to Phe 171 and Val 174 in *Drosophila* Orai, respectively) function as a dynamic 'gate' that prevents ion conduction when the channel is closed and permits ion permeation when the channel is open (*Frischauf et al., 2017*; *Yamashita et al., 2017*). The nature of the molecular rearrangements that open the channel and the conformations of these hydrophobic residues in the open state have been queried through functional studies of mutants and through molecular dynamics simulations (*Derler et al., 2013*; *Frischauf et al., 2017*; *Yamashita et al., 2020*; *Yamashita et al., 2017*; *Yeung et al., 2018*; *Yeung et al., 2020*), but they have not yet been addressed by high-resolution structural studies of an open conformation of the channel.

In this study, we present a cryo-electron microscopy (cryo-EM) structure of Orai$_{H206A}$ in an open conformation at 3.3 Å resolution. The structure agrees with our previous low-resolution structure of Orai$_{H206A}$ and provides a near-atomic view of the amino acids that line the pore in an open state. The substantially improved resolution was enabled through use of Fab antibody fragments that bind to the extracellular side of the channel and served as fiducial markers in the cryo-EM analysis. Details of the open pore, including the positioning of Phe 171 and conformational changes throughout Orai, provide insights into the mechanisms of ion permeation and gating in the channel. Additionally, the structure reveals that the connections between the M1 and M2 helices form structured turrets that create an extracellular entrance to the pore, clarifying the relationship between transmembrane helices within Orai channel subunits and suggesting a physiological role for these ordered structures.

## Results

### Antibody development and characterization

The remarkable advances in structural biology enabled by cryo-EM require that individual protein complexes embedded in vitreous ice can be identified within micrographs and that their orientations can be accurately determined (*Cheng, 2015*). As a consequence, complexes greater than 200 kDa and/or those with distinctly recognizable shapes typically yield higher resolution cryo-EM structures than smaller complexes or those with fewer distinguishing features. The somewhat spherical overall shape of the Orai channel and its relatively low molecular weight (~144 kDa) make obtaining a high-resolution cryo-EM reconstruction difficult. We therefore pursued a structure of Orai$_{H206A}$ in complex with monoclonal Fab antibody fragments that increase mass and serve as fiducial markers for determining particle orientations.

The monoclonal antibody 19B5 was developed by immunizing mice with purified Orai protein and selecting antibodies that bound to the purified channel but not to denatured protein, reasoning that an antibody that binds to a structural epitope would be more useful for structural studies (*Figure 1—figure supplement 1*, Materials and methods) (*Hunte and Michel, 2002*). Using mutagenesis and fluorescence-detection size exclusion chromatography (*Kawate and Gouaux, 2006*) to map its binding epitope, we determined that 19B5 binds to the extracellular side of the channel in the loop connecting the M1 and M2 helices (Materials and methods). Taken together, these binding properties suggested that the M1–M2 connection adopts a structured conformation even though density for this loop was weak in the previous X-ray structures of Orai.

Having identified 19B5 as a candidate for structural analysis, we sought to investigate the effect of the antibody on the function of the channel. To do so we performed $Ca^{2+}$ influx measurements in mammalian cells that expressed $Orai_{H206A}$ or wild-type Orai and STIM components. The addition of 19B5 antibody had no discernable effect on $Ca^{2+}$ influx through the channels, even though it can bind to Orai in a cellular context (*Figure 1* and *Figure 1—figure supplement 1*). We conclude that 19B5 does not markedly inhibit $Ca^{2+}$ flux through $Orai_{H206A}$ or through wild-type Orai when activated by STIM.

## Cryo-EM structure determination of the $Orai_{H206A}$–Fab complex

Purified $Orai_{H206A}$ was reconstituted into amphipols and combined with an excess of the 19B5 Fab antibody fragment for single particle cryo-electron microscopy (cryo-EM) analysis (Materials and methods). Classification of the particle images revealed Orai–Fab complexes containing one, two, or three Fabs bound per channel, as well as some free Fab fragments (*Figure 2—figure supplements 1* and *2*). We generated ab initio 3D reconstructions from particles that contained channels and found that these reconstructions contained six Orai subunits regardless of the number of Fab molecules that were bound, which indicated that the channels were uniformly composed of six Orai subunits (*Figure 2—figure supplement 2*). The Fab molecules bound to the same epitope regardless of their stoichiometry with the channel (*Figure 2—figure supplement 2*). As expected from the biochemical analyses, this epitope was located on the channel's extracellular side (*Figure 2*).

Particles of Orai–Fab complexes that contained three Fabs yielded higher-resolution cryo-EM maps and were used for structure determination. On account of the antibodies, cryo-EM reconstructions displayed threefold (C3) symmetry even when symmetry was not imposed (*Figure 2—figure supplements 1* and *2*) and therefore C3 symmetry was incorporated during subsequent cryo-EM processing. Fourier shell correlation (FSC) curves indicate that the final 3D reconstruction is determined at 3.3 Å overall resolution (*Figure 2* and *Figure 2—figure supplements 1* and *3*). Local

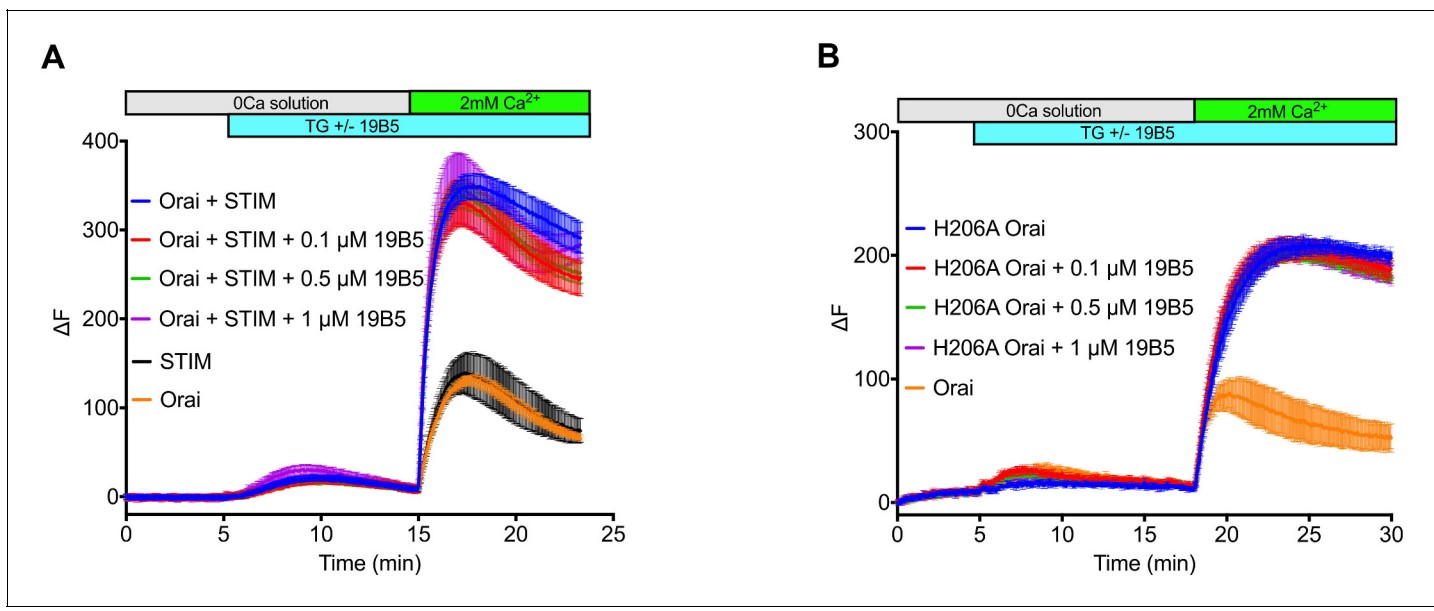

**Figure 1.** Analysis of the 19B5 antibody using a cellular $Ca^{2+}$ influx assay. $Ca^{2+}$ influx measurements were made from mammalian (HEK293) cells co-expressing wild-type *Drosophila* Orai with *Drosophila* STIM (A) or expressing the H206A mutant of Orai alone (B). Cytosolic $[Ca^{2+}]$ levels were detected using a genetically encoded fluorescent $Ca^{2+}$ indicator, GCaMP6s (*Chen et al., 2013*), as described previously (*Hou et al., 2018*). Data are plotted as the change in fluorescence intensity (ΔF). Indicted concentrations of 19B5 antibody were used. Thapsigargin (TG), 2 mM $CaCl_2$, and purified 19B5 antibody were added at the indicated times (horizontal bars). Controls (Orai or STIM alone) indicate background fluorescence levels. Standard error, derived from three independent measurements, is shown for each condition.

The online version of this article includes the following figure supplement(s) for figure 1:

**Figure supplement 1.** Further analyses of the 19B5 antibody.

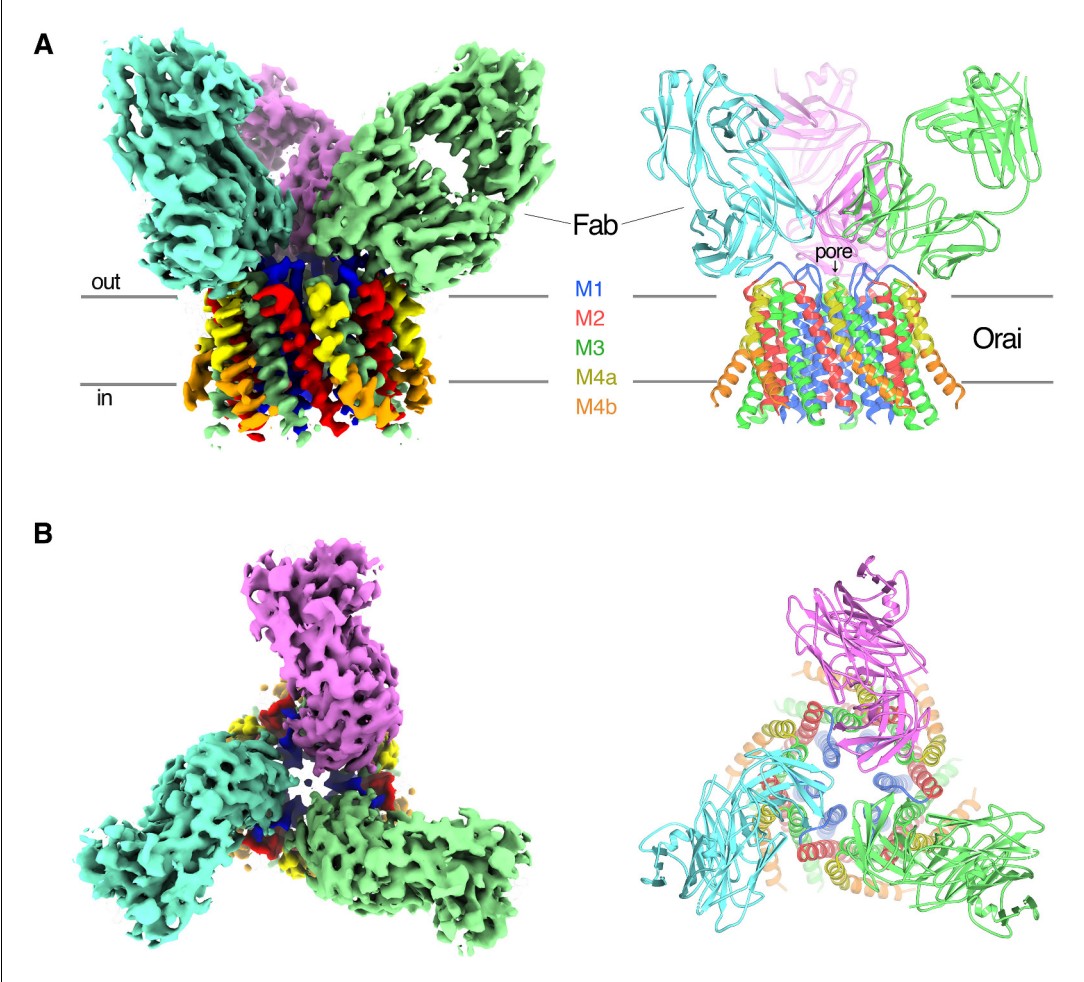

**Figure 2.** Orai$_{H206A}$–Fab complex. (A and B) Overall structure of the complex is shown from the perspective of the membrane (A) and from the extracellular side (B). The cryo-EM map (left) and the cartoon representation of the structure (right) are shown. Each Fab is colored individually; the α-helices of Orai are colored as indicated. Horizontal bars in (A) denote approximate boundaries of the plasma membrane.

The online version of this article includes the following source data and figure supplement(s) for figure 2:

**Source data 1.** Cryo-EM data collection, refinement, and validation statistics.
**Figure supplement 1.** Flowchart for cryo-EM data processing of the Orai$_{H206A}$–Fab complex.
**Figure supplement 2.** 3D reconstructions of Orai$_{H206A}$–Fab complexes containing between one and three Fab molecules.
**Figure supplement 3.** Cryo-EM structure determination and density.
**Figure supplement 4.** The Orai–Fab interface.

resolution estimates and visual inspection of the maps indicate that the central region of the complex comprising the M1, M2, and M3 helices and the variable domains of the Fabs have the most well-defined density (at ~3.1 Å resolution), while the M4 helices and the constant domains of the Fabs on the periphery are less well defined (*Figure 2—figure supplement 3*). The atomic model has good stereochemistry and good correlation with the cryo-EM density (*Figure 2—figure supplements 3* and *Figure 2—source data 1*). It contains the variable domains of three Fab molecules and amino acids 156–305 of Orai (except for the disordered M2–M3 loop, amino acids 217–239). The M4-ext helices (amino acids 306–341) and the N-terminal ends of the M1 helices (amino acids 144–156), which were observed in the X-ray structures, are not visible in the cryo-EM map, possibly due to flexibility in these regions and/or their peripheral locations in the cryo-EM reconstruction.

Each Fab binds to the extracellular side of Orai, adjacent to the entrance of the pore (*Figure 2* and *Figure 2—figure supplement 4*). There are six identical binding sites for the Fabs, on account of the hexameric architecture of Orai. While there are no visible contacts between the

antibodies, the channel can only accommodate up to three Fabs because of steric restraints (*Figure 2*). In accord with the $Ca^{2+}$ influx results (*Figure 1*), there is adequate room for $Ca^{2+}$ entry with Fabs bound (*Figure 2*). Each antibody primarily interacts with the connection between the M1 and M2 helices (*Figure 2—figure supplement 4*). This connection, which was not visible in previous X-ray structures of Orai and comprises amino acids 179–189, forms a structured 'turret' (*Figure 3*). Six turrets, from the six subunits, constitute the extracellular entrance of the pore (as described below). Each Fab predominately interacts with the M1–M2 turret of one subunit, but some contacts are also made with the turret of a neighboring subunit (*Figure 2* and *Figure 2—figure supplement 4*). These two adjacent turrets adopt indistinguishable conformations (*Figure 2—figure supplement 3*). This correspondence, in spite of the different interactions between the Fab and the two turrets, suggests that the antibody has minimal effects on the structure of the turret. The previously mentioned property that the antibody preferentially binds to the intact channel rather than to denatured protein provides further evidence that the turret is a structured element of the channel. Reinspection of the electron density from our previous X-ray studies of Orai (*Hou et al., 2018*; *Hou et al., 2012*) reveals weak density for the turret in both closed and open conformations (*Figure 4*). Although this density was not strong enough to direct model building, the density corroborates the conclusion that the turrets are structurally ordered.

The overall structure of the channel agrees with the low-resolution X-ray structure of Orai$_{H206A}$ within the error of that structure (*Figure 5A*). This correspondence of 3D structures determined using different methodologies and under different conditions (detergent micelles were used for the

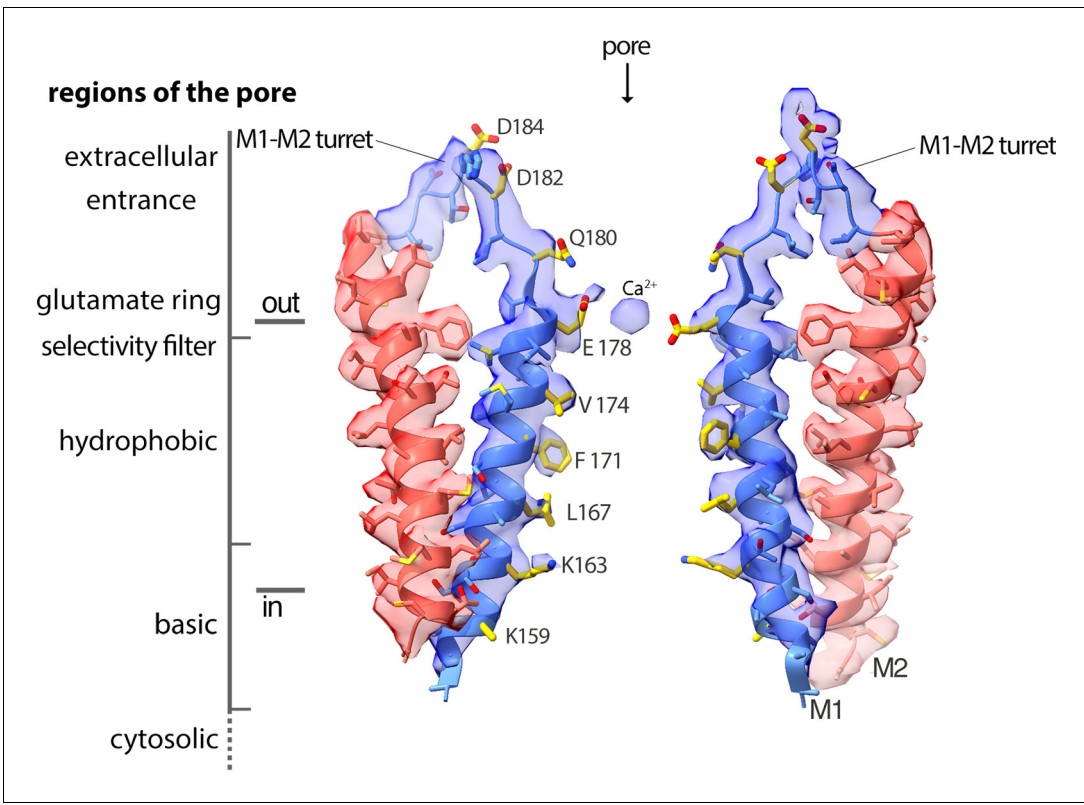

**Figure 3.** Cryo-EM density and structure of the pore. Cryo-EM density is displayed as semitransparent surface, showing the M1 through M2 portions of the channel from two opposing subunits (M3–M4 and other subunits are omitted for clarity). The atomic model is shown in cartoon representation, with amino acid side chains drawn as sticks. Amino acid side chains on M1 and on the M1–M2 turret that face the ion conduction pathway have yellow colored carbon atoms. Nitrogen and oxygen atoms are colored dark blue and red, respectively. Regions of the pore are indicated.

The online version of this article includes the following figure supplement(s) for figure 3:

**Figure supplement 1.** Density and approximate dimensions of the selectivity filter.

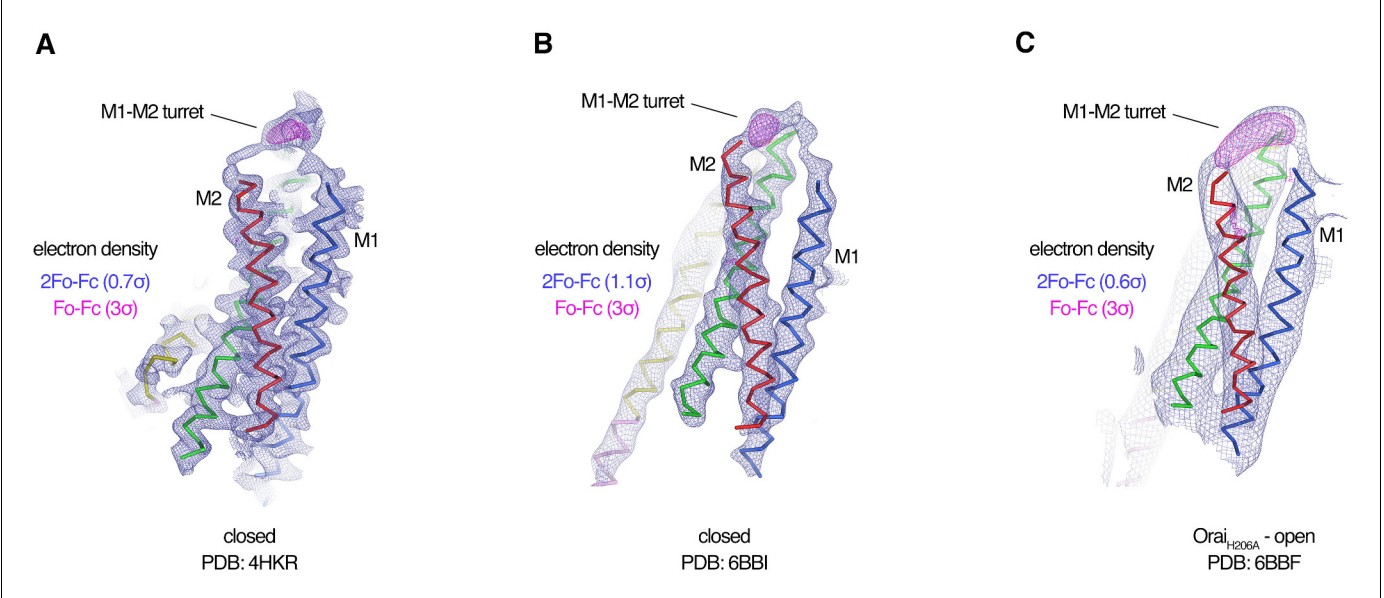

**Figure 4.** Evidence for the M1–M2 turret in previous X-ray structures of Orai. (A–C) An Orai subunit (Cα representation) is shown from three X-ray structures. In two of the structures (A and B), the pore of Orai is closed (*Hou et al., 2018*; *Hou et al., 2012*). The depiction in (C) is of the low-resolution X-ray structure of Orai$_{H206A}$ (*Hou et al., 2018*). 2Fo-Fc and Fo-Fc density maps are drawn as blue and magenta mesh, respectively, at the indicated σ levels. The electron density maps were calculated using the map coefficients FWT/PHWT and DELFWT/DELPHWT, respectively, that can be downloaded from the RCSB Protein Data Bank using PDB IDs: 4HKR, 6BBI, and 6BBF, respectively. In (C), 24-fold averaging was applied to the map in real space according to the non-crystallographic symmetry present in this crystal form to increase the signal-to-noise level. Weak 2Fo-Fc density for the M1–M2 turret is visible in each of the three structures (at 0.7σ, 1.1σ, and 0.6σ, respectively). Positive densities in the Fo-Fc difference maps at 3σ confirm the presence of the M1–M2 turrets even though they were not built in the models due to relatively weak density and/or low resolution X-ray data. The structures in (A and B) are of Orai in a quiescent conformation and of K163W Orai in an unlatched-closed conformation, respectively. In those structures, the pores adopt indistinguishable closed conformations (the M4-ext helices have different conformations) (*Hou et al., 2018*; *Hou et al., 2012*).

X-ray structure) and the abilities of the antibody to bind both wild-type and H206A Orai are further indications that the antibody binds to a native structure of the channel.

## Regions of the pore

With the additional appreciation of the structured extracellular turrets, the pore has five distinct sections (*Figures 3* and *6A*). From the extracellular to the intracellular side, these are: the extracellular entrance, the selectivity filter, a hydrophobic section, a basic section, and a cytosolic section. Aside from residues on the turret, the walls of the pore are formed by the M1 helices. As such, the amino acid side chains emanating from M1 create the physiochemical environment for the majority of the pore. The conformations of most of the pore-lining side chains are resolved in the cryo-EM map, providing a detailed view of the open pore of Orai$_{H206A}$ (*Figure 3*). *Figure 6A* shows the approximate dimensions of the open pore and indicates the residues lining its walls. Changes are evident along the length of M1 in comparison to the closed conformation of the pore (*Figure 6B*). These changes increase the diameter of the pore along its entire length. The regions of the open pore are described in more detail below.

## The extracellular entrance

The six M1–M2 turrets, one from each subunit, constitute the extracellular entrance of the pore (*Figures 3* and *5*). The amino acid sequence of the M1–M2 turret and its length are highly conserved among Orai channels (*Figure 7*). The cryo-EM structure establishes that the M2 helix of a given subunit is located directly behind the M1 helix of the same subunit (*Figures 3* and *5A*). The polypeptide of the turret adopts an extended secondary structure that connects the C-terminal end of M1 (at Glu 178) with the N-terminal end of M2 (at Gly 190). The turrets extend approximately 20 Å above the selectivity filter at Glu 178 and have a shape somewhat like that of an inverted 'V'. The improved

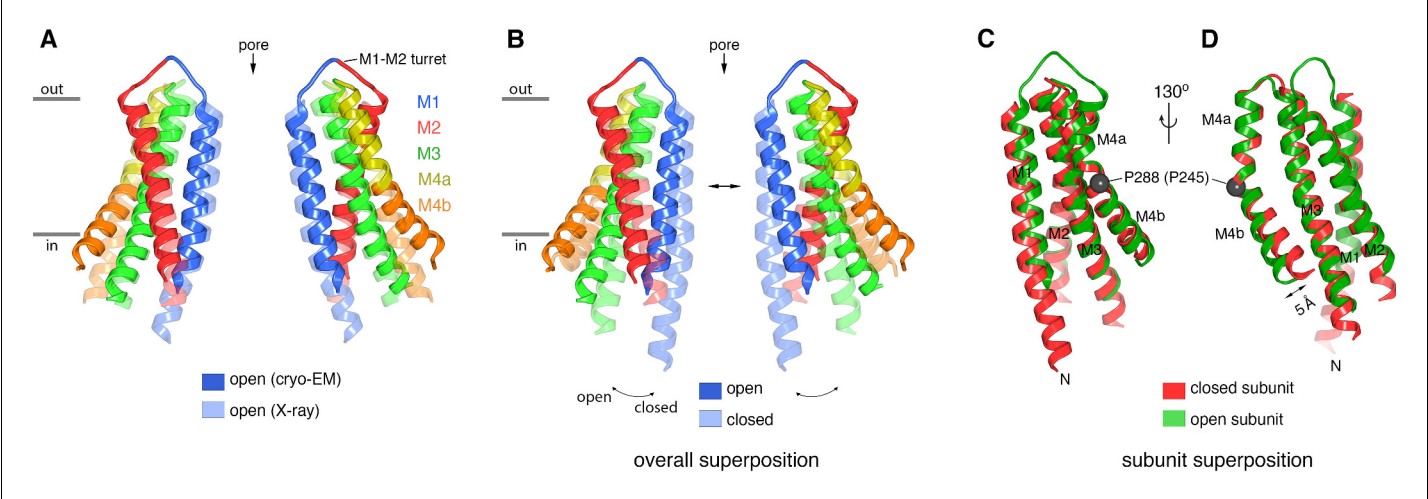

**Figure 5.** Comparison of the cryo-EM structure of Orai_{H206A} with other structures of the channel. (A) Comparison between the cryo-EM structure and a previous low-resolution X-ray structure of Orai_{H206A} (*Hou et al., 2018*). Two opposing subunits of Orai are depicted in cartoon representation and colored as indicated (four subunits are omitted for clarity). The X-ray structure (PDB: 6BBF) is colored in lighter shades. The M1–M2 turrets revealed by the cryo-EM structure, which create the extracellular entrance of the pore, are indicated. The mainchain RMSD between the structures is 1.5 Å. (B) Comparison between the cryo-EM structure of Orai_{H206A} and a 3.3 Å resolution X-ray structure of a closed conformation (quiescent conformation, PDB: 4HKR). The mainchain RMSD between these two structures is 2.7 Å. The depiction is as in (A), with the closed structure drawn in lighter shades. Arrows highlight dilation of the pore and the tilting of subunits. Superpositions in (A and B) were made by aligning complete hexameric channels. (C and D) Superposition of an individual subunit from Orai_{H206A} with an individual subunit of the closed structure (PDB: 4HKR), shown in two orientations (C and D). The mainchain RMSD between these two subunits is 0.74 Å. Aside from a slight displacement of the C-terminal portion of M4b (arrow), the conformations of an isolated subunit are highly similar between the open and closed structures.

resolution of this region indicates that the C-terminal end of the M1 α-helix is partially stabilized by hydrogen bonds between backbone oxygen atoms (at Met 176 and Val 177) and Lys 270 from the M3 helix of a neighboring subunit (*Figure 7D*). Amino acid sequence conservation of Lys 270 and of the surrounding region suggest that the corresponding residue in human Orai1, Lys 178, participates in similar interactions (*Figure 7E*).

The N-terminal end of the turret contains a conserved 'VQLD' motif (Val 179, Gln 180, Leu 181, and Asp 182) that is located immediately after Glu 178 of the selectivity filter (*Figure 7A–C*). The two hydrophilic residues of the motif, Gln 180 and Asp 182, are oriented toward the pore and the aqueous environment of the extracellular entrance. The two hydrophobic amino acids, Val 179 and Leu 181, point away from the aqueous environment and participate in a network of hydrophobic interactions that appear to provide structural stability to the turret (*Figure 7A and B*).

The extracellular entrance of the pore created by the turrets is considerably larger than was appreciated from the previous structures of Orai (*Figure 6*). The entrance is funnel shaped, with a diameter of ~20 Å at its widest point. It narrows to join with the selectivity filter of the channel at Glu 178. Gln 180, Asp 182, and Asp 184 contribute to the walls of the funnel, and together with Glu 178, these amino acids produce a markedly negative electrostatic surface that would tend to increase the local concentration of cations near the extracellular entrance of pore (*Figure 8*).

## The selectivity filter

A 'glutamate ring' of Glu 178 residues from the six subunits is the narrowest constriction of the open pore (*Figure 6A and C*). Previous functional and structural data indicate that the glutamate ring forms the selectivity filter of the channel that is responsible for the channel's high selectivity for $Ca^{2+}$ (*Hou et al., 2012*; *Prakriya et al., 2006*; *Vig et al., 2006*; *Yeromin et al., 2006*). Accordingly, density consistent with $Ca^{2+}$ is present at the center of the glutamate ring (*Figure 3*). The density corresponds to the binding location of $Ba^{2+}$, a permeable surrogate for $Ca^{2+}$, that was identified from our previous X-ray studies of Orai_{H206A} (*Hou et al., 2018*).

Cryo-EM densities for the side chains of the Glu 178 residues are relatively weak in comparison to other amino acids on the M1 helices, as is often the case for negatively charged amino acids in cryo-

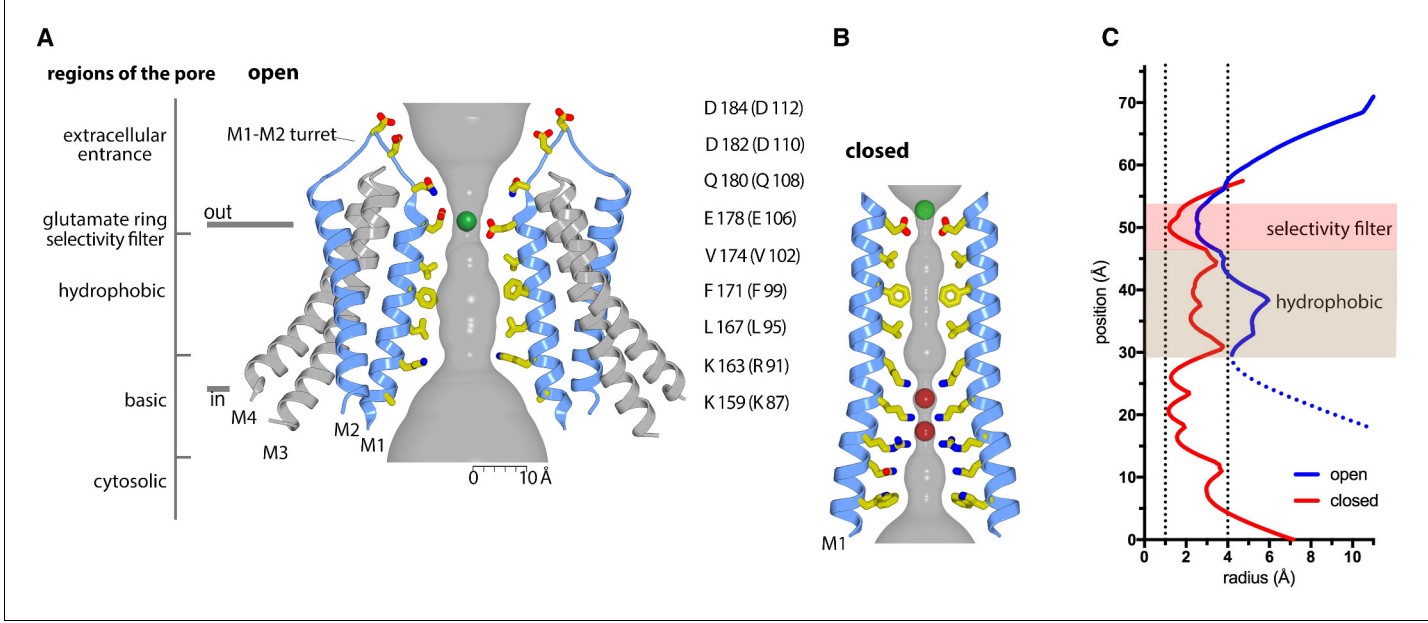

**Figure 6.** Ion pore and gating transitions. (A) Open pore of Orai_{H206A}. Two opposing subunits are shown as cartoons around the ion pore, which is depicted as a gray surface and represents the minimal radial distance to the nearest van der Waals contact. Amino acids lining the pore are shown as sticks (yellow carbon, red oxygen, and blue nitrogen atoms). These amino acids are labeled (the side chain of K159 is only partially modeled due to weak density); parentheses denote human Orai1 counterparts. A green sphere indicates $Ca^{2+}$. (B) Closed pore of Orai. Two opposing M1 helices from the X-ray structure of the quiescent conformation (PDB: 4HKR) are shown (ribbons) with amino acids that line the pore drawn as sticks. The ion pore is depicted as in (A); red spheres represent iron/anion binding sites. (C) Comparison of the dimensions of the closed and open pores. Pore radius (x-axis) denotes the minimal radial distance to the nearest van der Waals contact along the axis of the pore (y-axis). The positions and scale along the y-axis correspond to (A and B). The selectivity filter and hydrophobic regions are shaded. The blue trace indicates the radius for the final structure. Additional traces shown in *Figure 6—figure supplement 1* indicate the pore dimensions when all six Glu 178 residues are modeled in up or down conformations. The dotted line denotes uncertainty in the basic region due to weaker cryo-EM density. The ionic radius for dehydrated $Ca^{2+}$ (1.0 Å) and the radius of a hydrated $Ca^{2+}$ ion (approximately 4 Å) are indicated as vertical dashed lines.

The online version of this article includes the following figure supplement(s) for figure 6:

**Figure supplement 1.** Estimation of the minimal and maximal dimensions of the selectivity filter in the open structure.

EM structures owing to interactions with the electron beam (*Figure 3*, *Figure 2—figure supplement 3A*, and *Figure 3—figure supplement 1A and B*). Nevertheless, the weak densities hint that the side chains of the Glu 178 residues adopt two favorable rotamer conformations, with one glutamate residue in an 'up' rotamer and an adjacent glutamate in a 'down' rotamer, such that the amino acids alternate in an up–down manner around the hexameric pore (*Figure 3* and *Figure 2—figure supplement 3*). Structural modeling analyses suggest that each individual Glu 178 side chain could adopt an up or a down conformation without steric interference from the neighboring amino acid. Nevertheless, the observed alternating pattern suggest that this may be an energetically preferred configuration. The correlation with the C3 symmetry of the antibody binding suggests that the up–down pattern may be subtly biased by antibody binding, but the reason for this is not apparent from the structure since the antibodies are more than 20 Å from the glutamate ring. *Figure 3—figure supplement 1* indicates the distances between the modeled side chains and the calcium ion. However, we hasten to add that because of the weak density for the side chains, there is uncertainty associated with their positions. We qualitatively assessed this uncertainty by calculating the diameter of the filter when all six glutamates are modeled in up or in down conformations (*Figure 6—figure supplement 1*). For these reasons, it is important to avoid over interpretation of the coordination geometry in the structure and whether the interactions with $Ca^{2+}$ are direct or are mediated by water. These issues will be better addressed by future structural studies of an open channel at higher resolution.

In the closed conformation of the pore observed in the X-ray structure of wild-type Orai, all six of the Glu 178 residues are modeled in a down rotamer conformation (*Figure 6B*; *Hou et al., 2012*). However, there is a degree of ambiguity regarding the conformations of these residues in the closed

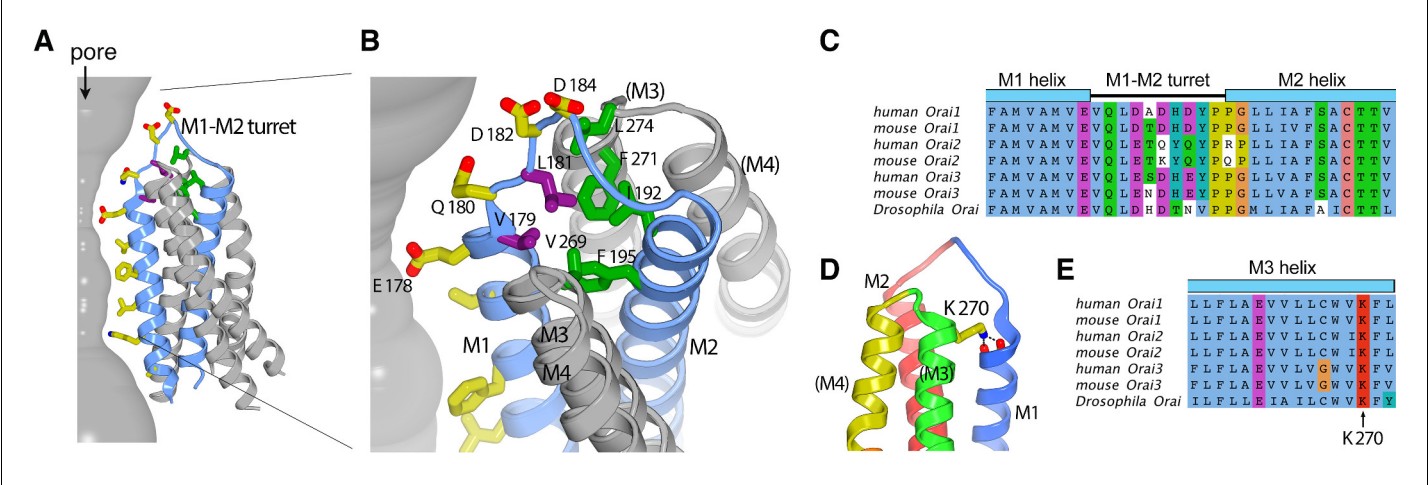

**Figure 7.** The M1–M2 turret. (A and B) Overall and close-up views highlighting the M1–M2 turret. The M1 and M2 helices and the intervening turret from a single subunit are light blue. The M3 and M4 helices from the corresponding subunit and a neighboring one are colored gray, with parentheses indicating helices from the neighboring subunit. Select residues are drawn as sticks: pore-lining amino acids (yellow carbon atoms), hydrophobic residues on the turret (purple), and cluster of hydrophobic residues beneath the turret (green). A section of the pore is depicted as a gray surface. (C) Sequence alignment in the turret region. Clustal coloring; portions of the M1 and M2 helices nearest the turret are shown. (D) Close-up view showing an interaction between Lys 270 and the C-terminal end of M1. Lys 270 is drawn as sticks; hydrogen bonds with the backbone carbonyl oxygen atoms at the C-terminal end of M1 are depicted as dashed lines. Parentheses indicate that the M3 and M4 helices are from a neighboring subunit. (E) Sequence alignment near Lys 270.

conformation as well. We hypothesize that the side chains are relatively dynamic and do not adopt a particular conformation for an extended period of time. Supporting this hypothesis, there are no additional in between the glutamate residues and other amino acids that might stabilize their conformations. This is in marked contrast to another highly selective $Ca^{2+}$ channel, the mitochondrial $Ca^{2+}$ uniporter, in which the conformations of a ring of glutamate residues in its selectivity filter are stabilized by van der Waals interactions and hydrogen bonds with other amino acids (*Baradaran et al., 2018*; *Fan et al., 2018*; *Fan et al., 2020*; *Nguyen et al., 2018*; *Wang et al., 2020*; *Wang et al., 2019*; *Yoo et al., 2018*).

In spite of some ambiguity regarding the conformations of the glutamate side chains in the selectivity filter of Orai, it is clear that the helical backbone of each M1 helix at Glu 178 is shifted away

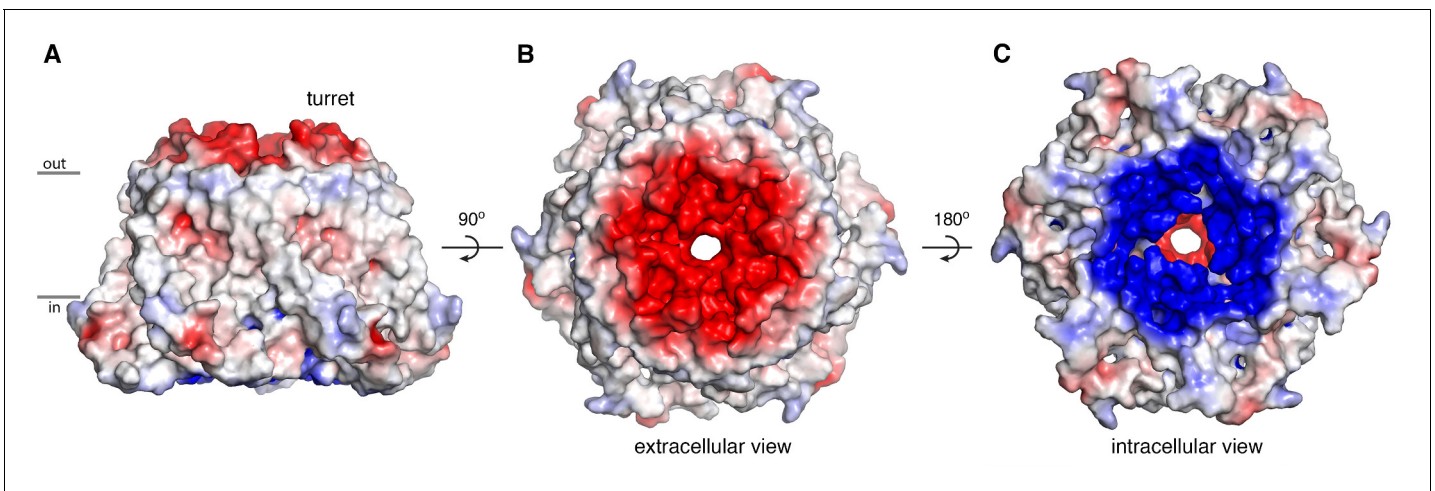

**Figure 8.** Electrostatic surface. (A–C) Molecular surface of Orai_H206A viewed from the membrane (A), extracellular (B), and intracellular (C) perspectives. Coloring is according to the electrostatic potential, which is contoured from –5 kT (red) to +5 kT (blue) (dielectric constant: 80).

from the central axis of the pore by approximately 1 Å relative to the closed conformation (*Figures 5B* and *6*). Thus, opening of the channel involves a degree of dilation within the selectivity filter.

## The hydrophobic region

Three perfectly conserved residues that are positioned on successive helical turns of each M1 helix, Leu 167, Phe 171, and Val 174 (Leu 95, Phe 99, and Val 102 in human Orai1), comprise a hydrophobic region of the pore (*Figure 6*). The entire hydrophobic region is markedly wider in the open conformation than when the pore is closed. Relative to the closed conformation, the helical backbone of each M1 helix is displaced by more than 2 Å away from the center of the pore in the open structure (*Figure 6* and *Figure 9*).

Changes within the hydrophobic region at and around Phe 171 are particularly noteworthy. In the closed conformation, the side chains of the six Phe 171 residues (from the six subunits) are in close proximity and are located near the central axis of the pore (*Figure 9B* and *Figure 9—figure supplement 1*). The conformation of this ring of phenylalanine residues is stabilized by hydrophobic packing among these residues and by close van der Waals contacts with the ring of Gly 170 residues behind them (*Figure 9B*). The packing is such that a Phe 171 residue of one subunit interacts with the Gly 170 residue of a neighboring subunit. The positioning of the Phe 171 residues shield the Gly

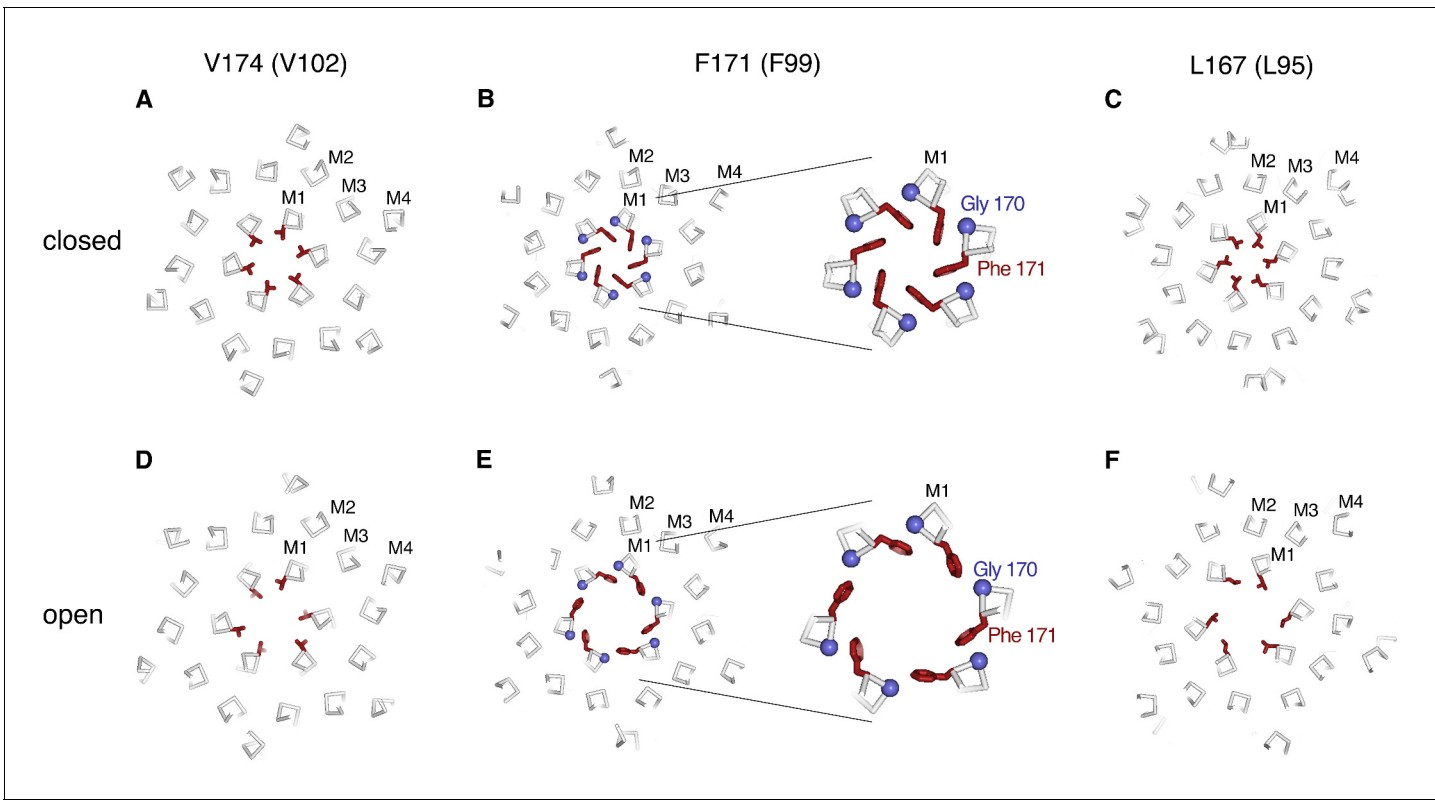

**Figure 9.** Conformational changes in the hydrophobic gate. (A–F) Slices through the hydrophobic region of the pore from the closed (PDB 4HKR) and Orai$_{H206A}$ open structures (upper and lower panels, respectively). Close up views in panels B and E highlight changes in the dimensions of the pore near Phe 171. The structures are represented as ribbons with amino acids side chains in the hydrophobic region drawn as sticks (dark red). Gly 170 is depicted as a blue sphere. The perspectives are from the extracellular side, perpendicular to the pore, which is centrally located. The slices correspond to approximately 4 Å slabs centered at Val 174, Phe 171, and Leu 167, as indicated. The M1–M4 helices of a single subunit are labeled. Inspection of the upper and lower panels indicates the outward rigid body movements of subunits. *Videos 1* and *2* depict these conformational changes.

The online version of this article includes the following figure supplement(s) for figure 9:

**Figure supplement 1.** Hydrophobic region.

**Figure supplement 2.** Residue 206 in the closed and open conformations.

**Figure supplement 3.** Analyses of P288L Orai.

170 residues from the pore in the closed state (*Figure 9B*). Opening of the channel increases the exposure of Gly 170 to the pore by an outward 'sliding' movement of Phe 171 that results from outward movements of the subunits (*Figure 9E*). Other than the increased exposure of Gly 170 to the pore in the open state, all of the amino acids that line the walls of the pore in the closed conformation also do so in the open conformation. Notably, the opening transition does not involve appreciable rotation of the M1 helices. Rather, rigid body movement of the helices away from the center of the pore constitutes the opening transition (*Figure 5B and C*, *Figure 6*, *Figure 9*, and *Videos 1* and *2*).

## The basic region

Oddly for a cation channel, the pore of Orai contains a highly basic region. Located just below the hydrophobic region, this basic region comprises 18 lysine or arginine residues (three residues from each of the six subunits) (*Figure 6*). These residues (Arg 155, Lys 159, and Lys 163 in *Drosophila* Orai) are conserved as lysine or arginine residues in all Orai channels (the corresponding amino acids are Arg 83, Lys 87, and Arg 91 in human Orai1). In the closed conformation of the pore, the basic amino acids are in close proximity and form binding sites for anions (*Hou et al., 2012*). Although the identity of the physiological anion is not yet established, X-ray and mass-spectrometry analyses indicate that purified wild-type Orai contains an iron complex within the basic region site that may be in the form of $(FeCl_6)^{-3}$ (*Hou et al., 2012*). The basic region is markedly wider in the open conformation owing to the outward movement of the M1 helices; the Cα positions of Lys 159 on opposing M1 helices are 12 Å further apart than in the closed structure (*Figure 6*). As was the case in the X-ray structure of Orai$_{H206A}$ (*Hou et al., 2018*), density for the anion/iron complex is not present within the widened basic region in the cryo-EM structure. Although side chain densities for the basic residues are less well defined than for other residues on M1, which suggests their flexibility, the helical register of M1 and some side-chain density for Lys 163 (*Figure 3*) indicate that the basic residues are exposed to the pore in the open state. We hypothesize that small cellular anions such as chloride would shield the basic amino acids from $Ca^{2+}$ ions permeating through the open channel.

## Gating conformational changes in the channel

The changes between the closed and open conformations of the channel are best described as rigid-body shifts in which the M1–M4 portion of each of the six subunits moves away from the center of the pore (*Figures 5*, *6,* and *9* and *Videos 1* and *2*). For instance, an ~2.5 Å outward movement of the M1 helix at Phe 171 within the hydrophobic region is accompanied by outward movements of similar magnitude for the M2, M3, and M4 helices (~1.6 Å,~2.0 Å, and ~1.7 Å measured at residues within the same horizontal plane as Phe 171, respectively) (*Figure 9B and E*). The outward movement is more dramatic on the cytosolic side of the channel and this results in slight

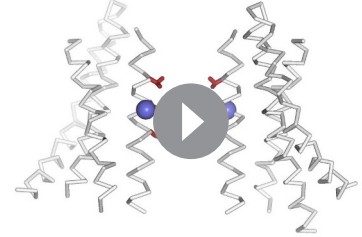

**Video 2.** Opening transition, showing a side view of the pore. Depicted similar to *Figure 9*, this video shows a side view of a morph between the closed and open conformations. Two opposite subunits are drawn as ribbons with amino acids in the hydrophobic region of the pore depicted as red sticks. Gly 170 residues are shown as blue spheres.
https://elifesciences.org/articles/62772#video2

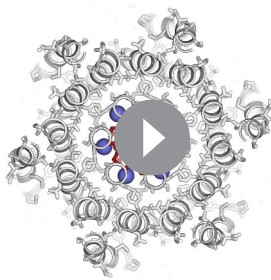

**Video 1.** Opening transition, depicted as in *Figure 9*, showing a morph between the closed and open conformations.
https://elifesciences.org/articles/62772#video1

tilting of subunits away from the pore on that side (*Figure 5B* and *Video 2*). In addition to these rigid body motions, a bend in the M4 helix at Pro 288, which delineates it into M4a and M4b, is less bent in the cryo-EM structure of Orai$_{H206A}$ than it is in the quiescent conformation. This results in ~5 Å outward movement of each of the M4b helices from its position in the quiescent conformation (measured at Ser 303) (*Figure 5B and D*).

His 206, the amino acid that when mutated to alanine gives rise to the constitutively activated channel, is located on M2 and would not contribute to the walls of the ion pore in either the closed or the open conformation (*Figure 9—figure supplement 2*). In the wild-type channel, His 206 makes a hydrogen bond with Ser 165 from M1 and is within a network of van der Waals interactions that involves residues from M1, M2, and M3 (*Frischauf et al., 2017*; *Hou et al., 2018*; *Hou et al., 2012*; *Yeung et al., 2018*). The H206A mutation eliminates this hydrogen bond, diminishes the van der Waals interactions (*Figure 9—figure supplement 2c*), and evidently alters the free energy profile of the channel sufficiently to favor an open state in the absence of STIM. A histidine can be readily modeled back into the Orai$_{H206A}$ structure, which suggests that the wild-type channel would be able to adopt the observed conformation (e.g. when activated by STIM) (*Figure 9—figure supplement 2b*).

Remarkably, the opening observed in Orai$_{H206A}$ structure does not involve notable side chain conformational changes, either in the pore or in the remainder of the channel. This can be appreciated by comparing amino acid conformations in *Figure 9* and the opening transition depicted in *Videos 1* and *2*. Because the subunits of Orai do not interdigitate and the $\alpha$ helices are nearly perpendicular to the membrane, we predict that the movement of an individual subunit would be relatively independent of neighboring subunits.

## Discussion

The cryo-EM structure of Orai$_{H206A}$ provides near-atomic detail of an open conformation of the channel. Somewhat surprisingly, opening of the pore does not involve twisting or bending movements of transmembrane helices that would be analogous to gating movements of many other cation channels such as voltage-dependent potassium, sodium, or calcium channels, as was first exemplified by structural studies of potassium channels (*Jiang et al., 2002b*; *MacKinnon, 2003*), nor does it involve marked rotation along helical axes. Rather, opening observed in Orai$_{H206A}$ involves rigid-body outward movement of each subunit. The amino acids within the transmembrane region adopt similar conformations in the closed and the open conformations (*Video 1*). The outward movement of the subunits results in a dilation of the pore and repositioning of Phe 171, which slides away from the central ion pathway to widen the hydrophobic region of the pore (*Figure 9* and *Videos 1* and *2*).

The open structure of the pore is in remarkable agreement with the amino acids predicted to contribute to the pore from cysteine accessibility experiments (*McNally et al., 2009*; *Yamashita et al., 2017*; *Zhou et al., 2010*). In spite of the potential disruptive effects caused by the introduction of cysteine mutations within a tightly packed and highly conserved ion pore, all of the residues that were predicted to contribute to the pore from these experiments (Arg 91, Leu 95, Gly 98, Phe 99, Val 102, and Glu 106 of human Orai1) are observed to do so in the structures. The accessibility profile of a cysteine substituted for Gly 98 (Gly 170 in *Drosophila* Orai) is particularly intriguing. In the background of the pore-lining V102A mutation, which is constitutively active (*McNally et al., 2012*) but hypothesized to be structurally similar to the closed conformation of the channel (*Yamashita et al., 2017*), channels bearing a cysteine substitution of Gly 98 were not blocked when $Cd^{2+}$ was applied to the extracellular side, but they were blocked when the mutant channel was activated by STIM (*Yamashita et al., 2017*). To account for this change in accessibility to $Cd^{2+}$, it was hypothesized that the M1 helix rotates modestly (by ~20°) along its helical axis when the channel opens (*Yamashita et al., 2017*). We do not observe a marked rotation of M1 between the closed and open structures, but we do observe a change in the exposure of the corresponding residue, Gly 170 (Gly 98 in Orai1), that could explain the observed accessibility profile. In the closed state, Gly 170 is hidden behind Phe 171 and is therefore not exposed to the pore (*Figure 9* and *Video 1*). However, Gly 170 becomes exposed to the pore in the open conformation due to the outward movements of the subunits and the sliding motion of Phe 171 relative to it (*Figure 9* and *Videos 1* and *2*). Yamashita et al. also observed a change in the ability of $Cd^{2+}$ to block the F99C mutant of Orai1 – the cysteine substitution of Phe 99 in Orai1 (corresponding to Phe 171 in Orai)

was blocked by $Cd^{2+}$ in the V102A closed state but was markedly less blocked in the STIM-activated state (*Yamashita et al., 2017*). This finding at first seems incongruous with the structures because Phe 171 is exposed to the pore in both the open and closed structures. However, a possible explanation for the accessibility profile of the F171C mutation is that the cysteine residues on opposing sides of the pore would be too far apart in the open conformation of the pore (~17 Å measured between sulfur atoms) to be coordinated by $Cd^{2+}$, which typically has a $S\text{-}Cd^{2+}$ coordination distance of ~2.5 Å and requires multiple sulfur ligands for efficient binding (*Rulísek and Vondrásek, 1998*).

In general agreement with our structural observations, molecular dynamics simulations of the H206A mutation have suggested that the hydrophobic region of the pore widens when the channel opens (*Frischauf et al., 2017*; *Yamashita et al., 2020*; *Yamashita et al., 2017*; *Yeung et al., 2018*). Although the details of the widening process (e.g. rotation vs. dilation) are slightly different between the cryo-EM structure and molecular dynamics simulations, the structure of $Orai_{H206A}$ supports the growing consensus that widening of the hydrophobic region underlies gating and it provides a structural basis for its mechanism (as discussed in more detail below).

Liu et al. recently described low resolution X-ray and cryo-EM structures of *Drosophila* Orai (determined at resolutions of ~4.5 Å and ~5.7 Å, respectively) with the mutation P288L (*Liu et al., 2019*), which has an activating phenotype in cells (*Liu et al., 2019*; *Nesin et al., 2014*; *Palty et al., 2015*). The authors propose that their structures reveal an open conformation of the pore (*Liu et al., 2019*), but we suspect that the pore is actually closed (non-conductive) in these structures. Side chain conformations are not visible owing to the low resolutions of these structures, but the densities for α-helices from both the cryo-EM and X-ray structures of P288L Orai reveal a conformation that is highly similar to an 'unlatched-closed' conformation that we observed previously (*Hou et al., 2018*) in which the pore is closed (*Figure 9—figure supplement 3b,c*). Most notably, the hydrophobic region is narrow in the P288L structure, as it is in the structures of Orai with a closed pore (*Hou et al., 2018*; *Hou et al., 2012*). Density consistent with an anion plug, which is a structural hallmark of the closed pore (*Hou et al., 2018*; *Hou et al., 2012*), is also observed in the P288L structures (*Figure 9—figure supplement 3c*). While the corresponding P245L mutation of human Orai1 causes activation of the channel when expressed in mammalian cells (*Liu et al., 2019*; *Nesin et al., 2014*; *Palty et al., 2015*), we have not been able to detect ion permeation through purified P288L Orai in liposomes, which suggests that the channel is not constitutively open on its own (*Figure 9—figure supplement 3a*). We suspect that channel activation by this mutation in cells may be dependent on STIM molecules that were endogenously present in the cell-based assays. Another gain-of-function mutation of human Orai1, T184M, has been reported to require STIM for its activation (*Böhm et al., 2017*). Further analysis of these intriguing mutations may shed light on the coupling between STIM and Orai and the mechanisms for how STIM activates the channel following $Ca^{2+}$ store depletion.

The M1–M2 turrets observed in the cryo-EM structure of $Orai_{H206A}$ reveal that the channel has an electronegative extracellular entrance that is wide enough to accommodate hydrated ions. The negatively charged residues on the walls of the entrance (Asp 182 and Asp 184, corresponding to Asp 110 and Asp 112 in human Orai1) would tend to concentrate $Ca^{2+}$ and other cations within vicinity of the pore, as has been proposed on the basis of functional studies and molecular dynamics simulations (*Frischauf et al., 2015*). In agreement with the locations of Asp 182 and Asp 184 in the structure, mutation of either residue to alanine has been shown to reduce the affinity of $Gd^{3+}$ block but not to effect $Ca^{2+}$ selectivity, which is mostly conferred by Glu 178 (*Prakriya et al., 2006*; *Vig et al., 2006*; *Yeromin et al., 2006*). (Glu 178, Asp 182, and Asp 184 of *Drosophila* Orai are referred to as Glu 180, Asp 184, and Asp 186, respectively, in *Yeromin et al., 2006*.) Experiments using cysteine substitutions in the M1–M2 loop of Orai1 reveal a pattern of accessibility to thiol-binding compounds (*McNally et al., 2009*) that suggests that the human channel has structurally similar M1–M2 turrets and an analogous extracellular entrance to that observed in the structure of *Drosophila* Orai. In those experiments, cysteine substitutions of the hydrophobic residues corresponding to Val 179 and Leu 181 in Orai (Val 107 and Leu 109 in Orai1), which are mostly buried in the structure, underwent modification by the cysteine-reactive compound MTSET (2-(trimethylammonium)ethyl methanethiosulfonate) approximately 10-fold more slowly than cysteine substitutions of the hydrophilic residues Gln 180 and Asp 182 (Gln 108 and Asp 110 in Orai1), which contribute to the solvent-exposed surface of the extracellular entrance in the structure. This cysteine accessibility profile suggests that the hydrophilic residues Gln 108 and Asp 110 of Orai1 are more exposed to the pore

than the hydrophobic ones (Val 107 and Leu 109 in Orai1), in accord with what we observe in the structure of the *Drosophila* channel. Through careful analysis of their data, and in excellent agreement with the structure of the extracellular entrance revealed by cryo-EM, the authors of that study concluded that the M1–M2 loops 'flank a wide outer vestibule' of the channel (*McNally et al., 2009*).

The cryo-EM structure indicates the M1 helices undergo approximately 2 Å outward movement at Glu 178 (between Cα atoms across the pore) that widens the selectivity filter when the pore opens. A widening of the selectivity filter is in agreement with several lines of evidence. Mutation of Val 102 in human Orai1 (Val 174 in Orai) to serine, cysteine, alanine, or other small amino acids creates channels that conduct cations without activation by STIM but which are considerably less selective for $Ca^{2+}$ (*McNally et al., 2012*). Ion selectivity for $Ca^{2+}$ is restored when STIM engages the mutant channel, which suggests that the opening of the channel by STIM induces a change in the selectivity filter (*McNally et al., 2012*). In additional evidence for a structural change in the filter, we have observed the repositioning of density for $Ba^{2+}$ within the filter between the X-ray structures of the closed channel and $Orai_{H206A}$ (*Hou et al., 2018*; *Hou et al., 2012*). $Ba^{2+}$, which is conductive in Orai, binds above the filter when the channel is closed, but appears to bind within the filter in the open pore. Density consistent with $Ca^{2+}$ in the cryo-EM structure presented here supports this repositioning upon opening (*Figure 6*). Finally, spectroscopic studies suggest that changes occur at the extracellular side of the pore when Orai opens (*Gudlur et al., 2014*). Thus, unlike classical potassium channels in which the selectivity filter can adopt a discrete conformation (to first approximation) regardless of whether the activation gate is open or closed (*Jiang et al., 2002a*; *Long et al., 2007*; *Zhou et al., 2001*), the selectivity filter of Orai adopts a discernably different conformation in an open conformation of the channel.

Owing to the relatively weak density for the Glu 178 side chains and the resolution limit of the reconstruction, the cryo-EM structure does not provide direct evidence regarding whether $Ca^{2+}$ coordination is direct or water mediated, nor does it directly address the number of $Ca^{2+}$ ions that can be accommodated in the filter. However, as we have described previously (*Hou et al., 2018*), we hypothesize that $Ca^{2+}$-selective ion permeation through Orai involves a mechanism, consistent with a classical understanding of $Ca^{2+}$ selectivity (*Sather and McCleskey, 2003*), in which two $Ca^{2+}$ ions are transiently positioned in single-file within the filter. In this model, the filter contains one $Ca^{2+}$ ion most of the time. This $Ca^{2+}$ ion blocks monovalent cation flux through the channel but it would not tend to leave the filter or permeate through the pore to a substantial degree due to a strong interaction with the surrounding glutamates of the filter. The entrance of a second $Ca^{2+}$ ion into the filter, however, could cause dissociation of the first due to repulsion between the ions while they both transiently occupy the filter. Flexibility of the glutamate side chains with a fraction of them in an up rotamer and a fraction in a down rotamer, as is suggested by the cryo-EM structure, may facilitate this transient accommodation of two $Ca^{2+}$ ions. This hypothesis is consistent with the observation that channels composed of concatemers of alternating wild-type human Orai1 subunits and subunits containing the E106Q mutation conduct cations but have reduced selectivity for $Ca^{2+}$ (*Cai et al., 2018*), whereas introducing this mutation into every subunit yields a non-conductive channel (*Prakriya et al., 2006*; *Vig et al., 2006*). While the current structure provides new insight into possible configurations of the glutamate ring, future structural studies at higher resolution and other experiments may enable a more complete understanding of the mechanism of ion selectivity.

As exemplified by the alternating conformations of the Glu 178 side chains, the cryo-EM structure suggests that there is fluidity in the conformations of amino acids on the pore such that the pore environment may not be perfectly symmetric at a given time. The conformations of the basic region may be particularly variable as density for this region is weaker than for other regions of the channel. We also hypothesize that the cytosolic region of M1, which is mostly disordered in the cryo-EM structure, may adopt a more defined conformation when the channel is activated by STIM, as is suggested by the ability of STIM and this portion of M1 to interact (*Derler et al., 2013*; *McNally et al., 2013*; *Zheng et al., 2013*).

The X-ray and cryo-EM structures of $Orai_{H206A}$ reveal that the basic region of the pore is markedly wider in the open conformation and no longer contains the anion plug that is observed when the pore is closed (*Hou et al., 2018*; *Hou et al., 2012*). Dilation of the basic region is consistent with previous biochemical experiments and molecular dynamics simulations that detected a widening of the pore in both the hydrophobic and basic regions in the activating H134A mutant of human Orai1

that corresponds to the H206A mutation of *Drosophila* Orai (*Frischauf et al., 2017*). However, several lines of evidence suggest that the basic region does not act as the primary impediment to ion permeation in the closed conformation of the channel. Richard Lewis and colleagues have shown that Orai1 channels in which any of the basic residues has been replaced by an acidic amino acid still form CRAC channels that require STIM for activation (*Mullins and Lewis, 2016*; *Mullins and Lewis, 2016*). Interestingly, such mutations eliminate $Ca^{2+}$-dependent inactivation, which suggests a role for the basic region in that process (*Mullins and Lewis, 2016*; *Mullins et al., 2016*). Further, we have shown that removal of the basic region by replacement of all three residues with serine (R155S, K159S, and K163S mutations) does not generate a constitutively active channel (*Hou et al., 2018*). Recent work by Murali Prakriya and colleagues showed that this triple serine substitution also prevents activation of the channel by STIM, in further support that the basic region is important for the ability of the channel to open and/or conduct $Ca^{2+}$ but suggesting that the basic region does not form the primary gate of the channel (*Yamashita et al., 2020*). We hypothesize that permeation of $Ca^{2+}$ through the basic region involves charge shielding by cellular anions such as $Cl^-$, as has also been proposed on the basis of molecular dynamics simulations (*Dong et al., 2013*). In accord with this hypothesis, our previous X-ray analyses have shown that $I^-$ can bind in the basic region of the H206A opened channel (*Hou et al., 2018*).

From the structures of Orai in open and closed conformations and the aforementioned functional and molecular dynamics studies (reviewed in *Yeung et al., 2020*), there is a consensus that the hydrophobic region of the pore functions as the primary impediment to ion permeation that controls $Ca^{2+}$ flow by operating as a 'gate'. In the closed conformation, the hydrophobic region is too narrow and too hydrophobic to permit $Ca^{2+}$ permeation, whereas the widened hydrophobic region in the Orai$_{H206A}$ structure would allow it. The hydrophobic nature of this region is particularly important to prevent ion conduction in the closed conformation: substitutions of hydrophobic amino acids within it with smaller or more hydrophilic ones (e.g. the V102A and F99C mutations of Orai1) allow cation permeation without activation by STIM (*McNally et al., 2012*; *Yamashita et al., 2017*). Molecular dynamics simulations of carbon-nanotube model systems and ion channels suggest that the ability of water molecules to be accommodated, even transiently, within hydrophobic regions is associated with their abilities to conduct ions (*Aryal et al., 2015*; *Jensen et al., 2010*). These studies suggest that when the diameter of a hydrophobic region is less than a certain width, the region becomes 'dewetted' in that water molecules are rarely observed in it, and ion permeation is prevented. Conversely, when the diameter is large enough, water molecules are accommodated and ion permeation is permitted. From simulations using a hydrophobic constriction similar in length to the hydrophobic section in the pore of Orai, a steep transition occurs at a diameter of approximately 9 Å (*Aryal et al., 2015*). When the diameter is 8 Å or less, the hydrophobic section is dewetted most of the time, whereas water molecules are observed within it at even slightly larger diameters. This corresponds well with the structures of Orai: the diameter of the hydrophobic region is 5–6 Å in the closed state and 9–10 Å in the H206A open state (*Figure 6*). The simulations also show that increasing the hydrophilicity of a constriction allows waters to populate it and that a hydrophilic constriction must be considerably narrower (<4 Å) than a hydrophobic one to prevent ion permeation (*Aryal et al., 2015*). These studies align with the observations that mutations of the hydrophobic region of Orai1 create leaky channels (e.g. V102A and F99C) and with molecular simulations that investigated the consequences of these mutations on pore hydration and conduction (*Dong et al., 2013*; *McNally et al., 2012*; *Yamashita et al., 2017*). While the data suggest that the hydrophobic region of the pore operates as the primary gate of the channel, we hasten to add that opening is concerted – the structures show that the pore of the channel widens along its entire length when it opens. This includes marked widening of both the hydrophobic and the basic regions. On account of the outward tilting of the subunits that underlies the opening observed for Orai$_{H206A}$, the widening is largest on the cytosolic side of the channel but it also extends to the extracellular side at the selectivity filter. As we have discussed, data indicate that the H206A mutation induces an open conformation that is highly similar to the STIM-activated wild-type channel. Nevertheless, there may be nuances of the STIM-activated pore that differ – a fuller understanding of the channel would be aided by structural information on a complex between Orai and STIM.

A theme has emerged from the studies of channels with different architectures that hydrophobic regions within their pores often function as gates that prevent ion permeation when these regions are narrow and permit it when they are widened (reviewed in *Aryal et al., 2015*). Different types of

molecular rearrangements engender the dilation and constriction in these 'hydrophobic gating' mechanisms. In the superfamily of cation channels that share a pore architecture first observed for a potassium channel, which includes voltage-dependent potassium, sodium, and calcium channels, a scissor-like movement of helices that also involves helical bending underlies the gating transition of a hydrophobic region of the pore (*Jiang et al., 2002b*; *MacKinnon, 2003*). A hydrophobic region in the chloride channel bestrophin opens and closes through a dramatically different mechanism that involves extensive side chain rearrangements throughout the channel (*Miller et al., 2019*). The transition in Orai is mechanistically simpler, involving the outward displacement of subunits through rigid body-like motions as observed in the Orai$_{H206A}$ structure. These examples of mechanisms for dimensional control of the hydrophobic gating regions align with the types of stimuli that control them. Voltage sensor domains in voltage-dependent potassium, sodium, and calcium channels are thought to squeeze shut the gates in these channels through mechanical coupling to the scissor-like conformational changes (*Jensen et al., 2012*; *Long et al., 2005*; *Wisedchaisri et al., 2019*; *Xu et al., 2019*). The molecular rearrangements of the gate in bestrophin channels are controlled allosterically by Ca$^{2+}$ and peptide binding to distant sites (*Vaisey and Long, 2018*; *Vaisey et al., 2016*). In Orai, the gate opens by outward movements of the subunits and is controlled by STIM from the cytosolic side, where these movements are largest. The structural mechanism for how STIM induces the change is an unresolved and intriguing issue. Our work highlights the utility of monoclonal antibodies as fiducial markers for determining high resolution cryo-EM structures of small proteins and provides a near-atomic framework to further investigate ion permeation, ion selectivity, and gating in CRAC channels.

## Materials and methods

### Key resources table

| Reagent type (species) or resource | Designation | Source or reference | Identifiers | Additional information |
|---|---|---|---|---|
| Gene *Drosophila melanogaster* | Orai | NCBI | Gene ID: 37040 | |
| Gene *Drosophila melanogaster* | STIM | NCBI | Gene ID: 32556 | |
| Recombinant DNA reagent | gCamp6s plasmid | *Chen et al., 2013* | | |
| Recombinant DNA reagent | pNEH-mCHerry plasmid | Modified from the pNGFP-EU vector (*Kawate and Gouaux, 2006*) | | |
| Cell line (*Homo sapiens*) | HEK293 | ATCC | CRL-1573, Lot #: 61714301 | |
| Antibody | *Drosophila* Orai-19B5 (mouse monoclonal) | Monoclonal Antibody Core Facility of the Memorial Sloan Kettering Cancer Center | | (1 μM for ELISA) |
| Antibody | Peroxidase AffiniPure Goat Anti-Mouse IgG, Fcγ fragment specific | Jackson Immuno Research | 115-035-071 | (1:2000 for ELISA) |
| Antibody | YL1/2 (rat monoclonal) | *Kilmartin et al., 1982* | | (20 mg per 1 g of CNBr resin) |

*Continued on next page*

*Continued*

| Reagent type (species) or resource | Designation | Source or reference | Identifiers | Additional information |
|---|---|---|---|---|
| Chemical compound, drug | 1-Palmitoyl-2-oleoyl-sn-glycero-3-phospho ethanolamine | Avanti Polar Lipids | 850757 | |
| Chemical compound, drug | 1-Palmitoyl-2-oleoyl-sn-glycero-3-phospho-(1'-rac-glycerol) (sodium salt) | Avanti Polar Lipids | 840457 | |
| Chemical compound, drug | n-Dodecyl-β-D-Maltopyranoside | Anatrace | O310S | |
| Chemical compound, drug | Amphipol A8-35 | Anatrace | A835 | |
| Chemical compound, drug | n-Octyl-β-D-maltopyranoside | Anatrace | O310 | |
| Chemical compound, drug | n-Decyl-β-D-maltopyranoside | Anatrace | D322 | |
| Chemical compound, drug | Deferoxamine mesylate | Sigma-Aldrich | D9533 | |
| Chemical compound, drug | Thapsigargin | Sigma-Aldrich | T9033 | |
| Chemical compound, drug | Lipofectamine 3000 Transfection Reagent | Thermo Fisher | L3000008 | |
| Chemical compound, drug | 1-Step Ultra TMB-ELISA Substrate | Thermo Scientific | 34029 | |
| Software, algorithm | MotionCor2 | *Zheng et al., 2017* | RRID:SCR_016499 | |
| Software, algorithm | CtfFind 4.1.10 | *Rohou and Grigorieff, 2015* | RRID:SCR_016731 | |
| Software, algorithm | RELION 3.0 | Scheres, 2016 | http://www2.mrc-lmb.cam.ac.uk/relion RRID:SCR_016274 | |
| Software, algorithm | SerialEM | Mastronarde, 2005 | RRID:SCR_017293 | |
| Software, algorithm | cryoSPARC v2 | Structura Biotechnology | https://cryosparc.com/ RRID:SCR_016501 | |
| Software, algorithm | PHENIX | *Liebschner et al., 2019* | https://www.phenix-online.org/ RRID:SCR_014224 | |
| Software, algorithm | COOT | *Emsley et al., 2010* | https://www2.mrc-lmb.cam.ac.uk/personal/pemsley/coot/ RRID:SCR_014222 | |
| Software, algorithm | PyMOL | Schrödinger, 2020 | https://pymol.org/2/ RRID:SCR_000305 | |

*Continued on next page*

*Continued*

| Reagent type (species) or resource | Designation | Source or reference | Identifiers | Additional information |
|---|---|---|---|---|
| Software, algorithm | UCSF Chimera | *Pettersen et al., 2004* | https://www.cgl.ucsf.edu/chimera RRID:SCR_004097 | |
| Software, algorithm | GraphPad Prism 7 | GraphPad Software | https://cryosparc.com/ RRID:SCR_016501 | |
| Software, algorithm | Hole | *Smart et al., 1996* | http://www.holeprogram.org | |
| Software, algorithm | UCSF ChimeraX | *Goddard et al., 2018* | https://www.cgl.ucsf.edu/chimerax/ | |
| Others | QUANTIFOIL R1.2/1.3 holey carbon grids | Quantifoil | | |
| Others | FEI Vitrobot Mark IV | FEI Thermo Fisher | | |

## Protein expression, purification, and cryo-EM sample preparation

The construct used to express Orai$_{H206A}$ is identical to the one previously used (*Hou et al., 2018*). This construct, spanning amino acids 133–341 of *Drosophila melanogaster* Orai, contains a C-terminal YL½-antibody affinity tag (EGEEF), mutations of two non-conserved cysteine residues that improve protein stability (C224S and C283T), and the H206A mutation. The expression of Orai$_{H206A}$ in *Pichia pastoris* and its purification were also as described (*Hou et al., 2018*) with minor modifications. Briefly, lysed cells were resuspended in buffer (8.3 ml of buffer for each 1 g of cells) containing 150 mM NaCl, 20 mM sodium phosphate, pH 7.5, 0.1 mg/ml deoxyribonuclease I (Sigma-Aldrich), 1:1000 dilution of Protease Inhibitor Cocktail Set III, EDTA free (CalBiochem), 1 mM benzamidine (Sigma-Aldrich), 0.5 mM 4-(2-aminoethyl) benzenesulfonyl fluoride hydrochloride (Gold Biotechnology), and 0.1 mg/ml soybean trypsin inhibitor (Sigma-Aldrich). Lysate was adjusted to pH 8.5 using 1 N KOH while stirring. Next, 0.1 g of n-dodecyl-β-D-maltopyranoside (DDM, Anatrace, solgrade) was added per gram of cells. The sample was stirred at 4°C for 1 hr to extract Orai$_{H206A}$ from the cell membranes. Following extraction, the pH of the sample was adjusted to 7.5 using 1 N KOH. The sample was centrifuged at 30,000 *g* for 45 min, and the sample supernatant was filtered using low-protein binding bottle-top filters (Millipore Express Plus 0.22 μm). This filtered supernatant was supplemented with final concentrations of 2 mM EDTA (stock EDTA 200 mM, pH 8.0), 2 mM EGTA (stock EGTA 200 mM, pH 7.0), and 0.2 mM deferoxamine mesylate (Sigma-Aldrich, 20 mM stock in water). YL½ antibody (IgG, expressed from hybridoma cells and purified by ion exchange chromatography) was coupled to CNBr-activated sepharose beads (GE Healthcare) according to the manufacturer's protocol. Approximately 0.2 ml of beads were added to the sample for each 1 g of *P. pastoris* cells and the mixture was rotated at 4°C for 1 hr. The beads were collected on a column and washed with seven column volumes of buffer consisting of 150 mM NaCl, 20 mM sodium phosphate, pH 7.5, 1 mM EDTA, 1 mM EGTA, 0.1 mM deferoxamine mesylate, and 3 mM DDM. Protein was eluted from the beads using a buffer consisting of 150 mM NaCl, 100 mM Tris, pH 8.5, 1 mM EDTA, 1 mM EGTA, 0.1 mM deferoxamine mesylate, 3 mM DDM, and 1 mM EEF peptide (Peptide 2.0). The elutant was concentrated to 500 μl (using an Amicon Ultra 15 100 kDa concentrator at 4°C), filtered using a 0.22 μm filter, and further purified using size exclusion chromatography (Superose 6 Increase 10/300 GL, GE Healthcare) at 4°C in SEC buffer (150 mM NaCl, 20 mM Tris, pH 8.5, and 1 mM DDM). Purified Orai$_{H206A}$ protein fractions were pooled and used to reconstitute into amphipols.

For reconstitution into amphipols, amphipol A8-35 (Anatrace, added as powder) was combined with the purified Orai$_{H206A}$ protein (12:1 wt/wt ratio, amphipol:protein) and the mixture was incubated at 4°C for 14 hr. Subsequently, approximately 0.333 g of wet Bio-Beads SM2 (Bio-Rad) were added to the sample for each milliliter of protein/amphipol mixture. After incubation at 4°C for 5 hr

(with rocking), the Bio-Beads were removed by spin filtration and the protein sample was concentrated to 500 µl using an Amicon Ultra 4 (100kDa) concentrator at 4°C. This amphipol-reconstituted sample was then purified by size exclusion chromatography (Superose 6 Increase 10/300 GL, GE Healthcare) at 4°C in buffer consisting of 150 mM NaCl, 20 mM Tris, pH 8.5, and 0.1 mM deferoxamine mesylate prior to combining it with purified 19B5 Fab.

## Fab production and complex preparation

A monoclonal antibody (designated 19B5) of isotype IgG1 was raised in mice by the Monoclonal Antibody Core Facility of the Memorial Sloan Kettering Cancer Center. The antigen used for immunization was that previously used for X-ray studies: purified Orai (amino acids 133–341 followed by the EEF affinity tag, PDB 6BBH) containing the K163W mutation, which improved protein stability, and has the same overall structure as channel without this mutation (*Hou et al., 2018*; *Hou et al., 2012*). The antibody selection process included ELISA, western blot, and FSEC analysis (*Kawate and Gouaux, 2006*) to identify antibodies that bound to native Orai and not SDS-denatured protein. For mapping its binding epitope, experiments included using a construct of Orai that contained an insertion of a Gly-Ala-Gly-Ala sequence in the M1–M2 loop combined with FSEC analysis, which indicated that it bound at or near this loop. We confirmed that 19B5 binds to the extracellular side of the channel by an ELISA-based pull-down experiment using intact HEK293 cells (ATCC CRL-1573, validated by the ATCC, tested negative for mycoplasma) that had been transfected with Orai or vector alone. The sequence of the antibody was determined by cDNA sequencing of hybridoma cells (SYD Labs). Intact IgG was expressed using mouse hybridoma cells, purified by ion exchange chromatography, and cleaved using papain (1:40 weight-to-weight ratio of papain [Worthington] to IgG) for 3 hr at 37°C to generate the Fab fragment. The Fab fragment was purified using ion exchange chromatography (Mono S, GE Healthcare; using a gradient of 10–500 mM NaCl in 20 mM sodium acetate, pH 5.0), dialyzed into 150 mM NaCl, 20 mM Tris-HCl, pH 8.5, and further purified using size exclusion chromatography (Superose 6 Increase 10/300 GL, GE Healthcare) in SEC buffer (150 mM NaCl, 20 mM Tris, pH 8.5, and 0.1 mM deferoxamine mesylate).

Prior to complex formation $CaCl_2$ was added to both purified $Orai_{H206A}$ and 19B5 Fab to 5 mM final concentration. The proteins were then combined at a molar ratio of 0.78:1 (Fab to Orai monomer, which corresponds to approximately 4.7 Fab molecules per channel assembly) and incubated on ice for 30 min. The sample was then concentrated to ~7 mg/ml (using a 10K Vivaspin two concentrator) and spin-filtered (0.22 µm, Costar). Three microliters of the sample was applied to glow-discharged (10 s) Quantfoil R1.2/1.3 holey carbon grids (Au 400, Electron Microscopy Sciences) and plunge-frozen in liquid ethane using a Vitrobot Mark IV (FEI) robot (settings: 6°C, blotting time of 2 s, 0 blot force, and 100% instrument humidity). Grids were stored in liquid nitrogen prior to data collection.

## Cryo-EM data acquisition

Clipped grids were loaded into a 300 keV Titan Krios microscope (FEI) equipped with a Gatan K2 Summit direct electron detector (Gatan). Grids were screened first for quality control based on particle distribution, particle density, and the estimated CTF resolution limit across grid regions. Images from the best regions of a single grid were collected at a magnification of 22,500× with a super-resolution pixel size of 0.5442 Å and a defocus range of −0.9 to −2.5 µm. The dose rate was nine electrons per physical pixel per second, and images were recorded for 10 s with 0.25 s subframes (40 total frames) corresponding to a total dose of approximately 76 electrons per $Å^2$.

## Cryo-EM data processing and structure determination

*Figure 2—figure supplement 1* represents the cryo-EM data processing workflow. Movie stacks were dark and gain reference corrected, and subjected to twofold Fourier cropping to a pixel size of 1.0884 Å. These images were motion corrected and dose weighted using MotionCor2 (*Zheng et al., 2017*). Contrast Transfer Functions for motion-corrected micrographs were estimated using CTFFIND4 (*Rohou and Grigorieff, 2015*). Micrographs were inspected manually; those with poor-quality features, such as obvious cracks or ice contamination, as well as those micrographs with estimated CTF fits worse than 5 Å were excluded. Out of a total of 4212 collected movies, 3902 passed these two curation criteria. From these micrographs a population of 2,151,623 particles were picked

using the Laplacian-of-Gaussian autopicking feature in RELION 3 (*Zivanov et al., 2018*) and extracted using a particle box size of 384 pixels. 2D classification was conducted in Cryosparc2 (*Punjani et al., 2017*) and classes clearly representing contaminants were excluded, resulting in retention of 1,851,514 particles (86% of the data). These particles were used as input for Cryosparc2 Ab initio 3D model generation, requesting four output models (*Figure 2—figure supplement 1*). Particles that yielded Ab initio 3D models that appeared to contain an Orai channel bound to one or more Fabs by visual inspection were subjected to additional rounds of ab initio 3D model generation, from which four distinct models that contained one to three Fab molecules per channel emerged (*Figure 2—figure supplements 1* and *2*). Particles containing three Fab molecules per channel yielded maps with the highest resolution, exhibited threefold rotational symmetry (C3), and were selected for all further cryo-EM processing steps. Following Cryosparc2 nonuniform refinement, the selected particles were further sorted by five iterations of heterogeneous 3D classification in Cryosparc2 (using C1 symmetry) to remove remaining assembles that contained fewer than three Fabs or poor particle images; this procedure yielded the final particle set of 85,614 particles. At this point, nonuniform refinement yielded a 3.7 Å reconstruction. The particles were then used for Bayesian polishing in RELION 3. Nonuniform refinement of the polished particles (in cryosparc2) was used to generate the final reconstruction at 3.3 Å. All resolution estimates are based on gold-standard FSC calculations.

## Model building and refinement

The atomic model of Orai$_{H206A}$ was manually built into the sharpened cryo-EM map and refined in real space using the COOT software (*Emsley et al., 2010*). Previous X-ray structures of Orai (PDB: 4HKR and 6BBH) were used as reference. Further refinement of the atomic model was carried out in PHENIX (*Adams et al., 2010*) using real-space refinement. The final model has good stereochemistry and good FSC with the map (*Figure 2—figure supplements 3e* and *Figure 2—source data 1*). Structural figures were prepared with Pymol (pymol.org), Chimera (*Pettersen et al., 2004*), and HOLE (*Smart et al., 1996*). Electrostatic calculations used the APBS (*Baker et al., 2001*) plugin in Pymol.

## Live cell Ca$^{2+}$ influx measurements

Orai constructs were expressed with N-terminal mCherry tags using a vector that was modified from the pNGFP-EU vector, as described previously (*Hou et al., 2018*; *Kawate and Gouaux, 2006*). Constructs for wild-type Orai (amino acids 120–351) for and full-length *Drosophila melanogaster* STIM (denoted as 'Orai' and 'STIM' in *Figure 1* and *Figure 1—figure supplement 1C and D*) have been reported previously (*Hou et al., 2018*). H206A mutation was introduced into the wild-type Orai construct (amino acids 120–351) by mutagenesis PCR and is referred to as 'H206A Orai'.

HEK293 cells were maintained in Dulbecco's Modified Eagle's medium (DMEM, the Media Preparation Core of MSKCC) supplemented with 10% fetal bovine serum (Gibco, catalog A31606-02). Approximately $1.5 \times 10^6$ cells were co-transfected (in a six-well dish) using the Lipofectamine 3000 transfection reagent (Invitrogen, catalog L3000-015) with 2 µg GCaMP6s plasmid (*Chen et al., 2013*) and plasmids of Orai and/or STIM constructs, for which the P3000 reagent was used along with Lipofectamine 3000 according to the manufacturer's protocol (Invitrogen). 0.15 µg Orai-mCherry plasmid and/or 0.6 µg STIM plasmid were used for transfection for samples shown in *Figure 1A* and *Figure 1—figure supplement 1C*. 0.05 µg of plasmid was transfected for H206A Orai and the controls shown in *Figure 1B* and *Figure 1—figure supplement 1D* because larger amounts of H206A Orai plasmid were toxic (presumably due to constitutive activation). Approximately 20 hr after transfection, the cells were trypsinized (Corning, catalog 25–053 Cl), resuspended in FluoroBrite DMEM (GIBCO, catalog A18967-01) supplemented with 2 mM L-glutamine (GIBCO, catalog 25030–081), and seeded to a 384-well plate (~$3 \times 10^4$ cells per well). Approximately 18 hr after seeding, the cells were gently rinsed once with 0Ca solution (10 mM HEPES pH 7.4, 150 mM NaCl, 4.5 mM KCl, 3 mM MgCl$_2$, 0.5 mM EGTA, and 10 mM D-glucose). After removing the rinse solution, 20 µl of 0Ca solution was added to each well and the cells were incubated for 30–40 min before fluorescence measurements. Fluorescence measurements were recorded using a Hamamatsu FDSS at Ex/Em = 480/527 nm every 2 s throughout the course of the experiment. After recording for 5 min, 10 µl of a 0Ca solution supplemented with thapsigargin (to yield a 1 µM final concentration) and a range of

concentrations of purified 19B5 Fab antibody (to yield 0–1 µM final concentration) were added. After a subsequent incubation (~10 min), 10 µl of $Ca^{2+}$-spike solution (10 mM HEPES pH 7.4, 146 mM NaCl, 4.5 mM KCl, 10 mM $CaCl_2$, and 10 mM D-glucose) was added to each well to yield a final $Ca^{2+}$ concentration of approximately 2 mM, and recordings were taken for another 8–12 min. The fluorescence traces were generated from an average ± SEM of fluorescence reading from three wells. Fluorescence intensity measurements (F) were plotted directly (*Figure 1—figure supplement 1C and D*) or as ΔF (*Figure 1*) on the Y-axis, where F is the measured fluorescence, F0 is the initial fluorescence value, and ΔF = F − F0.

To assess the ability of 19B5 to bind to the channel in a cellular setting, we performed an assay similar to an ELISA using live cells. HEK293 cells were grown, transfected, and resuspended in FluoroBrite DMEM with 2 mM L-glutamine as indicated above. Purified 19B5 antibody (1 µM final concentration; $IgG_1$) was added to 200 µl of transfected cells ($6.5 \times 10^6$ cells/ml; cell viability >90%) for 20 min at room temperature. The samples were then washed two times by pelleting ($900 \times g$) and resuspending in media (FluoroBrite DMEM). Following this, a horseradish peroxidase-conjugated secondary antibody (Jackson ImmunoResearch, catalog 115-035-071, used at 1:2000 dilution) was incubated with the cells at room temperature for 15 min in media. The cells were then washed five times by pelleting and resuspending in media to remove unbound antibody and then they were resuspended using 100 µl 1-Step Ultra TMB-ELISA Substrate (ThermoScientific, catalog 34029). Following 15 min of incubation, 100 µl of 2M sulfuric acid was added to stop the reaction. The sample was clarified by centrifugation ($900 \times g$), and the absorbance (450 nM) of the supernatant was read on a SpectraMax M5 fluorometer (Molecular Devices) using a standard ELISA assay protocol. Four replicates for each expression construct were performed.

FSEC was used to evaluate protein expression in cells that were used for $Ca^{2+}$ influx and live-cell ELISA assays. Transfected cells were solubilized in buffer (40 mM Tris pH 8.5, 150 mM NaCl, 10 mM DDM, and 1:500 dilution of Protease Inhibitor Cocktail Set III, EDTA free [CalBiochem]) at 4°C with gentle agitation for 1 hr. Cell lysates were clarified by centrifugation ($20,000 \times g$ at 4 °C for 1 hr) and the supernatants were injected onto a size exclusion column (Superdex 200 Increase 10/300 GL; GE healthcare) in buffer (10 mM Tris-HCl pH 8.5, 150 mM NaCl, and 1 mM DDM) and the fluorescence of mCherry (ex: 585 nm, em: 615 nm) of the eluate was monitored (*Figure 1—figure supplement 1f*).

## Reconstitution and flux assay

Orai constructs were purified and reconstituted into lipid vesicles using published procedures (*Hou et al., 2018*; *Hou et al., 2012*). The constructs (amino acids 133–341 of Orai with P288L, H206A, or V174A mutations) are identical to one another except for the indicated mutations (*Hou et al., 2018*; *Hou et al., 2012*). Briefly, a lipid mixture containing 15 mg/ml POPE (1-palmitoyl-2-oleoyl-sn-glycero-3-phosphoethanolamine) and 5 mg/ml POPG (1-palmitoyl-2-oleoyl-sn-glycero-3-phospho(1'-rac-glycerol)) was prepared in water and solubilized with 8% (w/vol) n-octyl-β-D-maltopyranoside (V174A and P288L) or 8% (w/vol) n-decyl-β-D-maltopyranoside (H206A). Purified Orai protein was mixed with the solubilized lipids to obtain a final protein concentration of 0.1 mg/ml and a lipid concentration of 10 mg/ml. Detergent was removed by dialysis (15 kDa molecular weight cutoff) at 4°C for 5–7 days against a reconstitution buffer containing 10 mM HEPES pH 7.0, 150 mM KCl, and 0.2 mM ethylene glycol tetraacetic acid (EGTA), with daily buffer exchanges and utilizing a total volume of 14 l of reconstitution buffer. The reconstituted sample was sonicated (~30 s), aliquoted, flash-frozen in liquid nitrogen, and stored at −80°C.

The fluorescence-based flux assay was performed as previously described (*Hou et al., 2018*; *Hou et al., 2012*). In brief, vesicles were rapidly thawed (using 37°C water bath), sonicated for 5 s, incubated at room temperature for 2–4 hr before use, and then diluted 100-fold into a flux assay buffer containing 150 mM n-methyl-D-glucamine (NMDG), 10 mM HEPES pH 7.0, 0.2 mM EGTA, 0.5 mg/ml bovine serum albumin and 0.2 µM 9-amino-6-chloro-2-methoxyacridine (ACMA, from a 2 mM stock in DMSO). Fluorescence intensity measurements were collected every 30 s on a SpectraMax M5 fluorometer using Softmax Pro five software (Molecular Devices; excitation and emission set to 410 nm and 490 nm, respectively). The proton ionophore carbonyl cyanide m-chlorophenyl hydrazine (CCCP, 1 µM from a 1 mM stock in DMSO) was added between the 150 and 180 s time points and the sample was mixed by pipette. The potassium ionophore valinomycin (2 nM from a 2 µM stock in

DMSO) was added near the end of the experiment to establish baseline fluorescence and confirm vesicle integrity.

## Acknowledgements

We thank Richard Hite and members of the Long laboratory for discussions, MJ de la Cruz of the Memorial Sloan Kettering Cancer Center Cryo-EM facility for help with data collection, and Frances Weis-Garcia and the staff of the Antibody and Bioresource core facility at Memorial Sloan Kettering Cancer Center. This work was supported by NIH grants R01GM094273 and R35GM131921 (to SBL) and a core facilities support grant to Memorial Sloan Kettering Cancer Center (P30CA008748).

## Additional information

### Funding

| Funder | Grant reference number | Author |
| --- | --- | --- |
| National Institute of General Medical Sciences | R01GM094273 | Stephen Barstow Long |
| National Institute of General Medical Sciences | R35GM131921 | Stephen Barstow Long |
| National Cancer Institute | P30CA008748 | Stephen Barstow Long |

The funders had no role in study design, data collection and interpretation, or the decision to submit the work for publication.

### Author contributions

Xiaowei Hou, Conceptualization, Data curation, Formal analysis, Supervision, Validation, Investigation, Visualization, Methodology, Writing - review and editing; Ian R Outhwaite, Conceptualization, Data curation, Formal analysis, Validation, Investigation, Visualization, Methodology, Writing - review and editing; Leanne Pedi, Data curation, Investigation, Methodology, Antibody development; Stephen Barstow Long, Conceptualization, Supervision, Funding acquisition, Validation, Investigation, Visualization, Methodology, Writing - original draft, Project administration, Writing - review and editing

### Author ORCIDs

Ian R Outhwaite (iD) http://orcid.org/0000-0003-2037-3261
Stephen Barstow Long (iD) https://orcid.org/0000-0002-8144-1398

### Decision letter and Author response

Decision letter https://doi.org/10.7554/eLife.62772.sa1
Author response https://doi.org/10.7554/eLife.62772.sa2

## Additional files

### Supplementary files

• Transparent reporting form

### Data availability

The cryo-EM map and the atomic model have been deposited in the Electron Microscopy Data Bank (EMDB) and in the Protein Data Bank (PDB) with accession numbers EMD-23002 and 7KR5, respectively.

The following datasets were generated:

| Author(s) | Year | Dataset title | Dataset URL | Database and Identifier |
|---|---|---|---|---|
| Hou X, Outhwaite IR, Pedi L, Long SB | 2020 | Cryo-EM structure of the CRAC channel Orai in an open conformation; H206A gain-of-function mutation in complex with an antibody | http://www.ebi.ac.uk/pdbe/entry/emdb/EMD-23002 | Electron Microscopy Data Bank, EMD-23002 |
| Hou X, Outhwaite IR, Pedi L, Long SB | 2020 | Cryo-EM structure of the CRAC channel Orai in an open conformation; H206A gain-of-function mutation in complex with an antibody | https://www.rcsb.org/structure/7KR5 | RCSB Protein Data Bank, 7KR5 |

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
