## [Decision Letter]

Thank you for submitting your article "Cryo-EM structure of the calcium release-activated calcium channel Orai in an open conformation" for consideration by *eLife*. Your article has been reviewed by three peer reviewers, including Randy B Stockbridge as the Reviewing Editor and Reviewer #1, and the evaluation has been overseen by Richard Aldrich as the Senior Editor. The following individuals involved in review of your submission have agreed to reveal their identity: Vera Y Moiseenkova-Bell (Reviewer #2); Joel Meyerson (Reviewer #3).

The reviewers have discussed the reviews with one another and the Reviewing Editor has drafted this decision to help you prepare a revised submission.

Summary:

The current manuscript from the Long lab describes a 3.3 Å cryo-EM structure of a constitutively open mutant of the store-operated calcium channel, Orai, in complex with a functionally neutral Fab fragment. The structure shows good agreement with the 2018 open-state X-ray crystallography structure but with much-improved visualization of most channel features. The current structure provides new structural insights: first, the previously unresolved "turret," an electrostatic funnel composed of extra-membrane M1-M2 linkages, second, confirms the pore-lining sidechains in the open conformation, and third, provides a structural rationale for the effects of the constitutively activating H206A mutant. Although this manuscript provides little new functional data, it engages with a very active Orai literature in the lengthy discussion.

The cryo-EM work is well done, and the quality of the data and completeness of analysis meets a very high standard. Given that the insights largely corroborate the 2018 crystal structure by the same team, the reviewers recommend that the manuscript be classified as a "Research Advance" to the 2018 paper, https://elifesciences.org/articles/36758. In addition, since one of the biggest motivations for solving a high-resolution structure was to resolve the calcium-binding glutamates (E178), the reviewers thought that the manuscript could be improved with a more extended analysis/discussion of the structural model put forth for this feature.

Essential revisions:

1) The authors will need to update their Abstract to correspond to the standard format for Research Advances. The Abstract should refer to the original *eLife* article: "Previously we showed that XXXX (author, year). Here we show that YYYYY." See http://elifesciences.org/content/4/e05033#abstract for an example Abstract.

2) The authors should prepare an additional figure showing the cryo EM density together with the model of the intact glutamate ring (all 6 protomers) and bound Ca^2+^, and add additional discussion of the glutamate ring to the text, in response to the questions described below.

3) Was the Ca^2+^ coordination sphere considered when the Glu rotamers were modelled? Does the Ca^2+^ coordination distance and geometry make sense when the sidechains are in the alternating up/down configuration?

4) In subsection “The open pore” the authors suggest that the glutamates may be dynamic and do not adopt a particular conformation for an extended period of time. Yet, Figure 7C presents a very straightforward pore radius profile. Could the authors clarify whether they imagine disorganized rotamer movement, or a more concerted up/down swap of 1, 3, and 5 with 2, 4, and 6? If the former, how would the spectrum of rotamer positions change the profile and radius of the selectivity filter compared with the alternating up/down model? Does it make sense to think of/visualize the radius of the selectivity filter having a radius range instead?

5) In the Discussion, the authors "suspect" that the selectivity filter has two Ca^2+^ ions positioned in single file. The authors should either elaborate on the rationale for presenting this idea, or remove it. Specific questions regarding this proposal include: Would a second Ca^2+^ positioned above the first be close enough to the "up" glutamates to be directly coordinated, would this be a water-mediated interaction, or would the binding sites overlap? The density of the Ca^2+^ is fairly spherical, and not smeared out like you might expect if there were multiple binding modes, or if the selectivity filter ever hosted two calciums simultaneously.

6) It might be expected that Fab binding to any particular subunit would be insensitive to the rotameric position of the Glu in the filter, leading to an averaging of the glutamate rotamers across adjacent subunits. But the maps do show distinct features for the alternating glutamates in this region. Could the authors comment on this?

7) Based on the comparison with the crystallography data, it is convincing that the Fab is binding a native state, and the pore is obviously not physically occluded. However, there is no positive evidence that the Fab is even binding the channel in the Figure 1 experiments. Do the authors know that Fab is actually bound in the context of the intact cell membrane?

---

## [Author Response]

Essential revisions:1) The authors will need to update their Abstract to correspond to the standard format for Research Advances. The Abstract should refer to the original eLife article: "Previously we showed that XXXX (author, year). Here we show that YYYYY." See http://elifesciences.org/content/4/e05033#abstract for an example Abstract.

Thank you for this suggestion. We have edited the Abstract for this format.

2) The authors should prepare an additional figure showing the cryo EM density together with the model of the intact glutamate ring (all 6 protomers) and bound Ca^2+^, and add additional discussion of the glutamate ring to the text, in response to the questions described below.

Thank you for this suggestion. We have made Figure 3—figure supplement 1 showing additional views of the density for the glutamate ring. We also expanded the Discussion about the selectivity filter in the text. As carefully described in the text, the density for the glutamate side chains is weak. While we agree that a better molecular understanding of the selectivity mechanism was one of the motivations for the work, we must be careful to not over interpret the data with regard to the conformations of the glutamate side chains, as is discussed in the manuscript.

3) Was the Ca^2+^ coordination sphere considered when the Glu rotamers were modelled? Does the Ca^2+^ coordination distance and geometry make sense when the sidechains are in the alternating up/down configuration?

Thank you. We have included geometry information with the new figures. We state in the revised text: “Figure 3—figure supplement 1 indicates the distances between the modeled side chains and the calcium ion. However, we hasten to add that because of the weak density for the side chains, there is uncertainty associated with their positions. We qualitatively assessed this uncertainty by calculating the diameter of the filter when all six glutamates are modeled in up or in down conformations (Figure 6—figure supplement 1). For these reasons, it is important to avoid over interpretation of the coordination geometry in the structure and whether the interactions with Ca^2+^ are direct or are mediated by water. These issues will be better addressed by future structural studies of an open channel at higher resolution.”

4) In subsection “The open pore” the authors suggest that the glutamates may be dynamic and do not adopt a particular conformation for an extended period of time. Yet, Figure 7C presents a very straightforward pore radius profile. Could the authors clarify whether they imagine disorganized rotamer movement, or a more concerted up/down swap of 1, 3, and 5 with 2, 4, and 6? If the former, how would the spectrum of rotamer positions change the profile and radius of the selectivity filter compared with the alternating up/down model? Does it make sense to think of/visualize the radius of the selectivity filter having a radius range instead?

Thank you for these comments. We have added Figure 6—figure supplement 1 to indicate that there is uncertainty associated with the pore diameter and we discuss this uncertainty in the revised text. We have also expanded the Discussion of the proposed selectivity mechanism. While it seems that the conformations of the glutamate side chains tend to be concerted from our structural analysis, because of the uncertainties mentioned, we do not wish to speculate further in this regard.

5) In the Discussion, the authors "suspect" that the selectivity filter has two Ca^2+^ ions positioned in single file. The authors should either elaborate on the rationale for presenting this idea, or remove it. Specific questions regarding this proposal include: Would a second Ca^2+^ positioned above the first be close enough to the "up" glutamates to be directly coordinated, would this be a water-mediated interaction, or would the binding sites overlap? The density of the Ca^2+^ is fairly spherical, and not smeared out like you might expect if there were multiple binding modes, or if the selectivity filter ever hosted two calciums simultaneously.

Thank you for these comments. The calcium selectivity hypothesis was proposed in our previous work (Hou et al., 2018), and we cite this more clearly in the revised manuscript. The revised manuscript also includes a slightly expanded Discussion of the proposed mechanism.

6) It might be expected that Fab binding to any particular subunit would be insensitive to the rotameric position of the Glu in the filter, leading to an averaging of the glutamate rotamers across adjacent subunits. But the maps do show distinct features for the alternating glutamates in this region. Could the authors comment on this?

Thank you. We now comment on this in the revised text.

7) Based on the comparison with the crystallography data, it is convincing that the Fab is binding a native state, and the pore is obviously not physically occluded. However, there is no positive evidence that the Fab is even binding the channel in the Figure 1 experiments. Do the authors know that Fab is actually bound in the context of the intact cell membrane?

Thank you. We now include an experiment showing that the 19B5 antibody binds to the channel in a cellular setting (Figure 1—figure supplement 1E).